# A time-resolved theory of information encoding in recurrent neural networks

**Rainer Engelken**[*]
Zuckerman Mind, Brain, Behavior Institute
Columbia University
New York, USA

**Sven Goedeke**
Neural Network Dynamics and Computation
Institute of Genetics
University of Bonn
Bonn, Germany

## Abstract

Information encoding in neural circuits depends on how well time-varying stimuli are encoded by neural populations. Slow neuronal timescales, noise and network chaos can compromise reliable and rapid population response to external stimuli. A dynamic balance of externally incoming currents by strong recurrent inhibition was previously proposed as a mechanism to accurately and robustly encode a time-varying stimulus in balanced networks of binary neurons, but a theory for recurrent rate networks was missing. Here, we develop a non-stationary dynamic mean-field theory that transparently explains how a tight balance of excitatory currents by recurrent inhibition improves information encoding. We demonstrate that the mutual information rate of a time-varying input increases linearly with the tightness of balance, both in the presence of additive noise and with recurrently generated chaotic network fluctuations. We corroborated our findings in numerical experiments and demonstrated that recurrent networks with positive firing rates trained to transmit a time-varying stimulus generically use recurrent inhibition to increase the information rate. We also found that networks trained to transmit multiple independent time-varying signals spontaneously form multiple local inhibitory clusters, one for each input channel. Our findings suggest that feedforward excitatory input and local recurrent inhibition–as observed in many biological circuits–is a generic circuit motif for encoding and transmitting time-varying information in recurrent neural circuits.

## 1 Introduction

How fast and reliable time-varying incoming stimuli are encoded in the population activity of recurrent neural networks constrains information transmission between local circuits and inter-areal communication. While the brain has to respond rapidly and reliably to changes in the world, several factors can drastically hamper information encoding in recurrent network models: the timescales of synapses can be slow, single neurons exhibit Poisson-like temporally irregular dynamics and dynamic instability of recurrent neural circuit dynamics can be a source of chaotic variability [1, 2, 3, 4, 5].

Excitation and inhibition in biological neural networks is usually conveyed by different types of neurons with a predominance of recurrent inhibitory feedback, a property known as 'inhibition dominance' [6, 7, 8, 9]. In networks with very strong recurrent inhibition, where recurrent weights are scaled as $1/\sqrt{N}$ with network size $N$, a dynamic balance arises between excitatory currents and recurrent inhibitory currents that was originally proposed as a robust mechanism to describe the emergence of asynchronous irregular activity [1]. Networks of binary neurons in such a *balanced state* track time-varying signals [2]. While subsequent experimental work found evidence for a dynamic

---

[*]Correspondence to re2365@columbia.edu

36th Conference on Neural Information Processing Systems (NeurIPS 2022).

balance of input currents, the 'tightness' or 'looseness' of such a dynamic balance in cortical circuits, as well as the computational implications, are a subject of active scientific debate [10]. 'Loosely' balanced networks, where excitatory and inhibitory currents are respectively $\mathcal{O}(1)$ compared to the distance from reset to threshold, can respond in their population firing rate nonlinearly to external inputs, which was argued to be favorable for sensory processing [7]. 'Tight balance' refers to a more precise tracking of total excitatory and total inhibitory input currents in time originating from excitatory and inhibitory currents that are respectively $\mathcal{O}(\sqrt{N})$ or larger compared to the distance from reset to threshold [10, 1, 2]. Such 'tight balance' was also studied in a series of works that arrived at spiking balanced networks from a normative predictive coding ansatz [11, 12, 13, 14]. How the dynamic tracking of time-dependent inputs described in binary networks [1, 2] extends to firing rate models and how 'tightness of balance' affects the information encoding of time-varying stimuli in the presence of chaos and noise has not yet been addressed. Previous dynamic mean-field theory (DMFT) approaches to input-driven rate networks assumed that the mean of the external inputs across neurons does not depend on time, which facilitates DMFT [15, 16, 17, 14], but prohibits to investigate encoding of dynamic stimuli in the population firing rate. To address this gap, we study how a time-varying stimulus is encoded in the population rate of an inhibition-dominated rate network under the influence of additive noise and chaos.

We show that the accuracy of neuronal population encoding improves in more tightly balanced recurrent networks because of a speedup of the effective timescale of the population mean dynamics. Conventional methods of dynamic mean-field theory [18, 19, 20, 17] are not adequate to capture the effects of a time-varying common input. Therefore, we use a dynamic mean-field theory that is non-stationary, meaning that the order parameters are time-dependent. Beyond a similar recent approach [21], our theory can treat both time-varying common and independent external input, which is crucial to analytically treat a signal-to-noise ratio that constrains population coding. This novel technique accurately captures the statistics of the input-driven networks. Specifically, we calculate the cross-spectral density between input and output and the power spectral density of the population firing rate. Together with the knowledge of the input statistics, this allows us to calculate the mutual information between stimulus and population rate in the Gaussian channel approximation based on the spectral coherence.

We examine how the frequency-response and mutual information rate depends on the tightness of balance, added noise, network chaos, and statistics of the input stimulus, using both theory and simulation. All the analytic results match those from network simulations. We show that recurrent networks that are trained on tracking a time-dependent stimulus develop strong recurrent inhibition and strong positive input weights, a fingerprint of the balanced state. Concomitantly, the mutual information rate between their network readout and the stimulus increases, as predicted by our theory. This indicates that a more tightly balanced state is a generic solution to reliably transmitting a time-varying input in the presence of noise or chaos. Lastly, we find that networks trained on simultaneously transmitting multiple independent time-varying stimuli spontaneously break up into weakly connected subnetworks with strong local inhibition.

Our findings have important implications for information encoding in RNNs and for understanding how neural network architecture design shapes noise-robustness and information encoding.

The main contributions are the following:

- A novel non-stationary mean-field theory describing the dynamics of balanced RNNs driven by a time-varying input signal and noise (section 2 and appendix A).

- A transparent mathematical link between tightness of balance and the mutual information rate between a scalar input signal and the population rate (section 4 and appendix D).

- A frequency-resolved mutual information rate analysis showing the effects of noise and chaos (section 3 and appendix C).

- Training results indicating that vanilla networks trained on transmitting high-frequency signals become more tightly balanced throughout training (section 6 and appendix E).

- Additional training results showing that networks trained to transmit multiple independent signal exhibit a spontaneous symmetry breaking into weakly connected subnetworks with strong local inhibition (section 7 and appendix I).

- Additional training results with a nonlinear task (section 8), different activation function (appendix H) and excitatory-inhibitory architecture (G) demonstrate generality of our results.

## 2 Population coding in recurrent networks

We study how well a time-varying scalar input $I(t)$ is encoded in the mean population firing rate $\nu(t) = \frac{1}{N}\sum_i \phi(h_i(t))$ of a recurrent network of $N$ nonlinear rate units ('neurons') that obey

$$\tau\frac{\mathrm{d}h_i}{\mathrm{d}t} = -h_i + \sum_{j=1}^{N} J_{ij}\phi\left(h_j\right) + bI(t) + \xi_i(t)\,, \tag{1}$$

with the coupling weights $J_{ij} = -bJ_0/N + \tilde{J}_{ij}$. The $\tilde{J}_{ij}$ are drawn independently from a Gaussian distribution with zero mean and variance $g^2/N$, where $g$ is a gain parameter that controls the heterogeneity of weights. We use a threshold-linear transfer function $\phi(x) = \max(x, 0)$. The

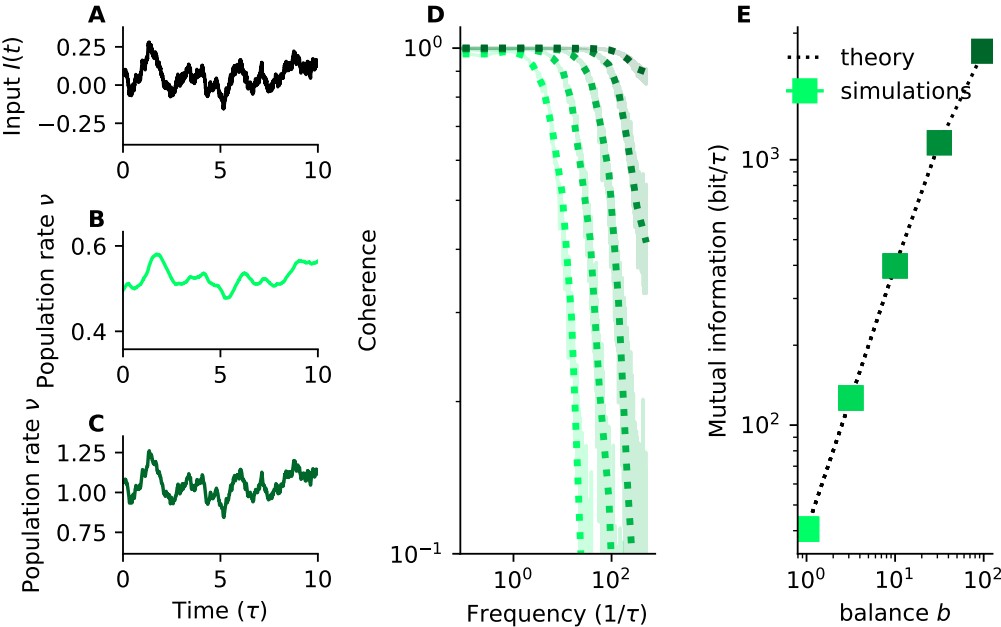

Figure 1: **Mutual information rate between input and population response grows with tightness of balance A)** Each neuron in the recurrent network receives an identical input stimulus $I(t)$ and additive Gaussian white noise $\xi_i(t)$ that is independent across neurons. **B)** For small values of $b$ ('loosely' balanced networks), the population response only tracks slow fluctuations of the stimulus ($b = 1$). **C)** For large values of $b$ ('tightly' balanced networks), the population response also tracks fast fluctuations of the stimulus ($b = 100$). **D)** The spectral coherence for different values of balance $b$, direct numerical simulations (shaded line) and mean-field theory (dashed line) superimposed. (For color-code, see points in figure 1E). **E)** The mutual information rate in Gaussian channel approximation based on spectral coherence, mean-field theory (dashed line) and direct numerical simulation (squares). Model parameters: $N = 2^{12}$, $g = 0$, $\Delta t = 1/2^{10}\tau$, $t_{\mathrm{sim}} = 2^{10}\tau$, $I_0 = J_0 = 1$, $\tau_S = \tau$, $\sigma = 0.1$, $I_1 = 1/2^3$, nperseg=$2^{13}$.

external input contains a signal component $I(t)$, which is identical across neurons, and a noise term $\xi_i(t)$, which is independent across neurons. For concreteness, we choose $\xi_i$ to be independent additive white Gaussian noise processes (AWGN) with autocorrelation function $\langle \xi_i(t)\xi_i(t+t')\rangle = \tau\sigma^2\delta(t')$ and the signal $I(t)$ to be an Ornstein-Uhlenbeck process given by

$$\tau_S\frac{\mathrm{d}I}{\mathrm{d}t}(t) = -I(t) + I_0 + \xi_S(t)\,. \tag{2}$$

The amplitude of the AWGN $\xi_S(t)$ driving the time-varying part is denoted by $I_1$. The parameter $b$ multiplies both external input and recurrent coupling strength and thus regulates the tightness of balance [14, 10, 21]. For firing-rate networks, the ability to track time-dependent input strongly

depends on the strength of balance $b$ (Fig 1). For loosely balanced network ($b = 1$) (Fig 1B), the network population rate only tracks slow fluctuations of the input signal. In contrast, for tightly balance networks (Fig 1C), the population firing rate also tracks fast input fluctuations. To understand how this tracking arises in the model, it is useful to rewrite Eq 1 by decomposing $h_i = m + \tilde{h}_i$ into a common component and residual fluctuations. With $J_{ij} = -bJ_0/N + \tilde{J}_{ij}$, this results in

$$\tau \frac{\mathrm{d}m}{\mathrm{d}t} = -m - bJ_0\nu(t) + bI(t)\,, \tag{3a}$$

$$\tau \frac{\mathrm{d}\tilde{h}_i}{\mathrm{d}t} = -\tilde{h}_i + \sum_j \tilde{J}_{ij}\phi\left(h_j\right) + \xi_i(t)\,. \tag{3b}$$

Here, the signal $I(t)$ enters the expression for $m$, because it is identical across all neurons. It thus directly impacts the mean population rate $\nu(t)$ and recruits, through the negative mean coupling $-bJ_0$, strong recurrent feedback $-bJ_0\nu(t)$ that is anti-correlated with the input and cancels most of the common external input $I(t)$. Rewriting Eq 3a by solving for $\nu(t)$ gives

$$\nu(t) = \frac{I(t)}{J_0} + \frac{1}{bJ_0}\left(-\tau\frac{\mathrm{d}m}{\mathrm{d}t} - m\right)\,. \tag{4}$$

This equation is commonly referred to as the 'balance equation' [2, 19, 20] when the input is not time-dependent. For large $b$, the population rate approaches $\nu(t) = I(t)/J_0$, which indicates a linear relationship between input and population rate for tightly balanced networks as pointed out before in binary networks [1, 2]. On the other hand, the noise $\xi_i$ affects only the residual fluctuations $\tilde{h}_i$.

As the balance parameter $b$ is increased, the effective timescale of the dynamics of $m$ becomes shorter by a factor of $b$, as can be seen by dividing Eq 3a by $b$ resulting in a timescale of $\tau/b$. This leads to faster and more precise tracking. In contrast, the timescale of the equation for $\tilde{h}_i$ is unchanged by $b$. Assuming a separation of timescales, for increasing $b$, the linearized rate dynamics develops a negative real outlier eigenvalue of size $-b$. Of course, the two equations for $m$ and $\tilde{h}_i$ are interdependent via $\phi(h_i) = \phi(m + \tilde{h}_i)$, so for a complete treatment, the joint dynamics of $m$ and $\tilde{h}_i$ has to be solved. In appendix A we present a non-stationary dynamic mean-field theory of temporal population coding.

## 3 Frequency-dependent signal encoding in the population response

We systematically analyze how different input frequencies are encoded in the population response and observe that the transmission of high frequency stimuli is improved for large $b$ ('tightly balanced' regime) (Figure 2A and C). We calculate the dynamic gain $G(f) = |S_{I\nu}|/|S_{II}|$ of the recurrent network both in direct numerical simulations and with the non-stationary DMFT. $S_{I\nu}$ is the Fourier transformation of the input-output cross-correlation function between $I(t)$ and $\nu(t)$. $S_{II}$ is the power spectral density of the stimulus $I(t)$. The dynamic gain quantifies how a variation in the input signal $I(t)$ affects the population rate $\nu(t)$ and is a standard analysis for characterizing the response properties of individual cells [22].

We calculate $G(f)$ using both direct numerical network simulations and dynamic mean-field theory. For details of the spectral analysis, see appendix C. We observe that increasing $b$ boosts the dynamic gain for high frequencies (Figure 2A and C for $g = 0$ and $g = 2$). This is a direct consequence of the speedup of the dynamics of the mean discussed above. Thus, as the network becomes more tightly balanced, high-frequency signals can be encoded and transmitted more reliably. To quantify the improved information encoding we turn to an information-theoretic analysis next.

## 4 Mutual information rate between time-varying stimulus and population response

With information theory, we can treat the recurrent neural network as a noisy communication channel, transforming a signal embedded in a noisy current into a population rate that can be read out by another population. We use a Gaussian channel approximation to the mutual information rate between input current $I(t)$ and output mean population rate $\nu(t)$ based on the spectral coherence [23, 24]:

$$R_{\mathrm{lb}}(I, \nu) = -\int_0^{f_{\mathrm{cutoff}}} df \log_2\left(1 - C_{I\nu}(f)\right)\,, \tag{5}$$

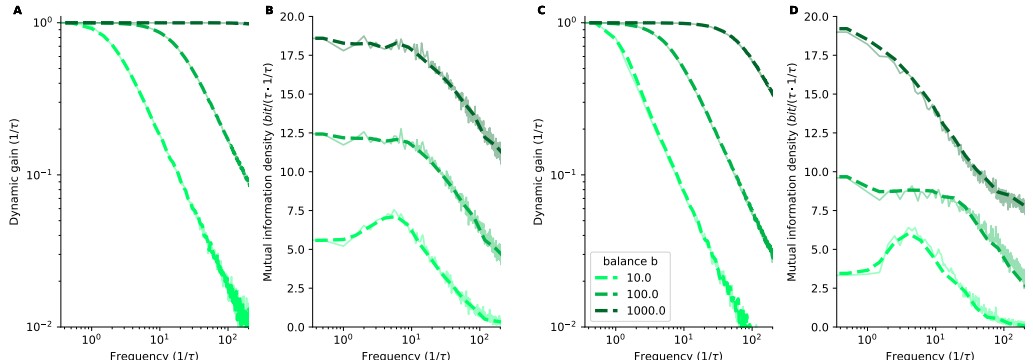

Figure 2: **Frequency-resolved information rate reveals different effects of white noise and chaos**
**A)** The normalized dynamic gain for different values of balance $b$, direct numerical simulations (shaded line) and mean-field theory (dashed line) superimposed for $g = 0$. Note that for large values of $b$, the dynamic gain shows an improved encoding bandwidth (For color-code, see figure 2C).
**B)** The mutual information rate density in the Gaussian channel approximation based on spectral coherence, mean-field theory (dashed line) and direct numerical simulation (transparent full lines) for $g = 0$. **C)** Same as **A)** but for $g = 2$. **D)** Same as **B)** but for $g = 2$. The recurrent residual fluctuations in $\tilde{h}_i$ arising from $g > 0$ reduce the information rate for low-frequency information. For larger values of $b$ ('tight balance'), recurrent fluctuations have weaker effect on low-frequency information rate, because of tracking of network fluctuations. Model parameters: $N = 4096$, $g = 0$ for **A)** and **B)**. $g = 2$ for **C)** and **D)**, $\Delta t = 1/2^{10}\tau$, $t_{\text{sim}} = 2^6\tau$, $I_0 = J_0 = 1$, $\tau_S = \tau$, $\sigma = 0.1$, $I_1 = 1/8$.

where $C_{I\nu}(f)$ denotes the magnitude squared spectral coherence. The spectral coherence is the frequency-domain analog of the correlation and measures the linear relationship between frequency components of input and output signal. Its magnitude square is

$$C_{I\nu}(f) = \frac{|S_{I\nu}(f)|^2}{S_{II}(f)S_{\nu\nu}(f)}. \tag{6}$$

It is normalized by the power spectrum $S_{II}(f)$ of the input signal (an OU process) and the power spectrum $S_{\nu\nu}(f)$ of the population rate $\nu(t)$.

We calculated the spectral coherence in two independent ways: firstly based on the non-stationary DMFT and secondly based on direct numerical simulations. The dependence of $C_{I\nu}(f)$ on the tightness of balance $b$ is depicted in Figure 1D. DMFT and direct numerical simulations with additive white noise $\xi(t)$ are in excellent agreement.

We find that the mutual information rate grows approximately linear with $b$ for an OU-input signal and Gaussian white noise (see Figure 1E and analytical approximation in appendix D). This is consistent with an analytical derivation of the mutual information rate in a linear approximation, where we find the mutual information rate (in $nat/\tau$) to be

$$R_{\text{lb}}(I, \nu) = \frac{bI_1\sqrt{N}}{2\sigma\sqrt{\tau_S/\tau}}, \tag{7}$$

for $bI_1\sqrt{\tau_S/\tau} \gg \sigma/\sqrt{N}$, i.e., for large networks and large signal-to-noise ratio. The mutual information rate saturates for large $b$; the saturation level is determined by the band limit of the incoming signal and noise (see figure in appendix C). We conclude that the mutual information rate scales approximately linearly with the tightness of balance $b$ for a sufficiently high band limit. This conclusion is not restricted to networks of rectified linear units, a similar reasoning holds also for the threshold-power-law nonlinearities and other neuron models (See e.g. [19, 20] for balanced rate networks with other input-output transfer functions). Finally, we note that for an input signal that is not an OU-process, we would expect a different scaling of the mutual information rate.

## 5 Frequency-resolved mutual information rate analysis

Inspired by recent work on the frequency-resolved information encoding rate in single cells [25], we analyze the population rate of RNNs and study how the encoding of different frequency components

of the input signal depends on the tightness of balance $b$, the strength of added noise $\sigma$ and the heterogeneity of the recurrent weights $g$.

For that, we considered the mutual information rate density $r(f)$, which gives a frequency-resolved quantification of the information encoding rate (See for details of definition [25]):

$$r_{\text{lb}}(f) = -\log_2\left(1 - C_{I\nu}(f)\right) \tag{8}$$

Note that since the integral over frequencies of $r_{\text{lb}}(f)$ gives the Gaussian channel approximation of the mutual information rate, we thus call it mutual information rate density and remark that it has units of $\frac{bit/\tau}{1/\tau}$. We observe that tightly balanced networks can transmit more high-frequency information (Figure 2B and D for both $g = 0$ and $g = 2$). Overall, the information rate for more tightly balanced networks is increased because of the boost of the signal-to-noise ratio coming from the fact that the signal $I(t)$ is scaled by $b$, while the noise is independent of $b$. The above-described frequency-dependent effect comes on top of that.

For recurrent networks with weight heterogeneity ($g > 0$), there are additional slow residual fluctuations in $\tilde{h}$ that act as an additional source of variability. Moreover, increasing $g$ leads to an increasing fraction of units in saturation, which reduces information transmission (Compare Figure 2B for $g = 0$ and D for $g = 2$). These low-frequency fluctuations have a different effect than additive Gaussian white noise (see also [14]).

For more tightly balanced networks (larger $b$), the residual fluctuations do not impair information encoding (see darker green lines in figure 2B), because they are canceled. A related phenomenon was observed in densely connected recurrent binary networks ([26] and spiking networks ([27]). That variance originating from weight heterogeneity ($g > 0$) is more effectively reduced by balanced networks as a function of $b$ compared to variance originating from added Gaussian white noise $\sigma$ was already observed in rate networks with static input [14].

## 6 Training networks on auto-encoding results in tight balance

We corroborate the theoretical finding that the tightness of balance has a crucial role in information encoding in numerical experiments in trained recurrent networks, which demonstrates the generality of our findings. For that, we train recurrent networks on an auto-encoder task (Fig 3). We initialize small-sized recurrent networks ($N = 100$) that follow the dynamics of Eq. 1 in a loosely balanced state, with recurrent weights drawn i.i.d. from a Gaussian $J_{ij} \sim \mathcal{N}(-1/N, g^2/N)$ and input weights $w_i^{\text{in}}$ and output weights $w_i^{\text{out}}$ are drawn from a Gaussian distribution with a positive mean. This initialization would correspond to $b = 1$ in our theoretically studied model (Eq. 1). The network receives the time-varying input signal $I_i(t) = w_i^{\text{in}}I(t)$, so in contrast to our theoretical setup, in principle each neuron can receive the input signal $I(t)$ with a different scale $w_i^{\text{in}}$. We then trained input weights $w^{\text{in}}$, output weights $w^{\text{out}}$ and recurrent weights $J_{ij}$ using backpropagation-through-time with the ADAM optimizer with standard hyperparameters (see appendix E for additional details on training setup and additional controls of training results). We minimize the mean squared loss $l = \int |\hat{I}(t) - I(t)|^2 \, dt$ between a time-varying input signal $I(t)$ and a linear readout of the recurrent network activity $\hat{I}(t) = 1/N \sum_i w_i^{\text{out}}\phi(h_i(t))$.

Initially, the network only poorly tracks the time-varying input signal, as theoretically expected for small $b$. However, we find that the network learns to track the time-varying input signal (Figure 3A), as the test error falls throughout training epochs (Figure 3D). Analyzing the eigenvalues of the trained network, we observe an outlier eigenvalue with a very negative real part after training that indicates an emerging strongly negative mode implementing the fast tracking (Figure 3B). This is confirmed by measuring the empirical tightness of balance $\hat{b}$ in the connectivity, which we quantify simply by the absolute value of the mean of the sum of incoming recurrent weights into each neuron $\hat{b} = |1/N \sum_i \sum_j J_{ij}|$. This definition of $\hat{b}$ is consistent with the definition of $b$ in our theory above. We find that the trained networks that are initialized with $b = 1$ become progressively more tightly balanced (Figure 3C) over training epochs, as can be seen by a growing $\hat{b}$ and by an improving high-frequency encoding (Figure 3E). We prematurely stop after 100000 training epochs to avoid numerical instabilities for very tightly balanced networks. We note that the average of the external input weights $\frac{1}{N} \sum_i w_i^{\text{in}}$ becomes more positive over training epochs (Figure 3C inset, green line)

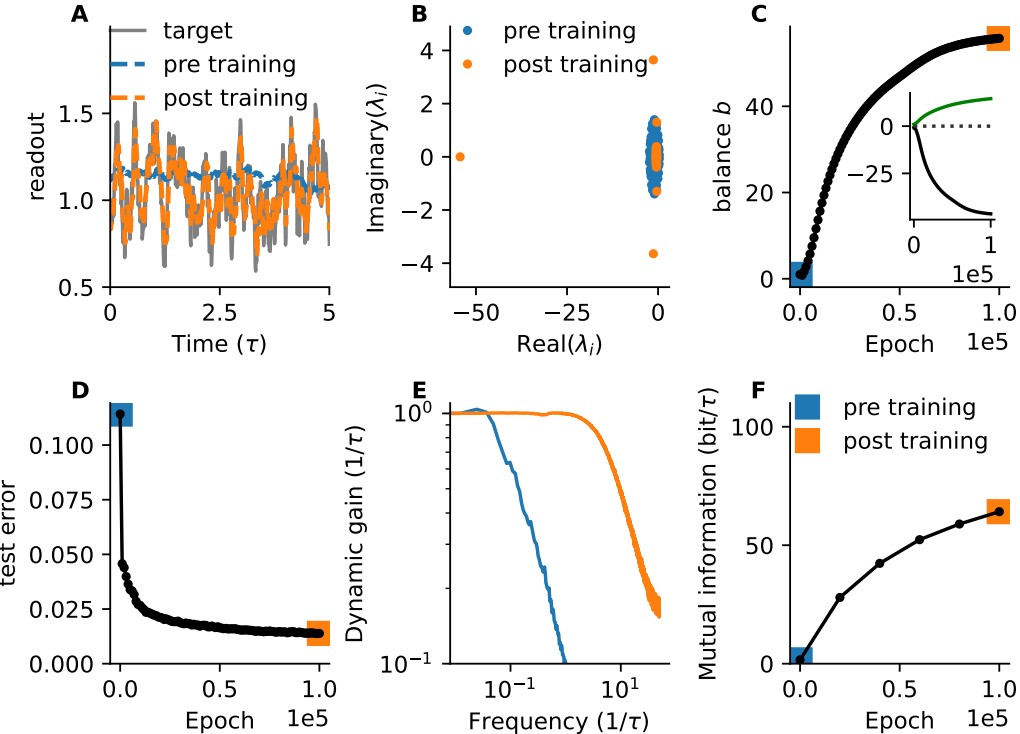

Figure 3: **RNNs trained on tracking time-varying input become more tightly balanced throughout training.** **A)** Threshold-linear RNNs are trained to approximate a time-varying external input $I(t)$ by linear readout $\hat{I}(t) = 1/N \sum_i w_i^{\text{out}} \phi(h_i)$, by minimizing the mean squared loss $l = \int |\hat{I}(t) - I(t)|^2 \, dt$. **B)** The eigenvalue spectrum of the dynamics linearized at the fixed point without time-dependent input indicates that during training, an eigenmode with a strongly negative real part emerges. **C)** The networks become more balanced throughout training. The tightness of balanced is quantified here by the magnitude of the mean recurrent coupling $\hat{b} = |1/N \sum_i \sum_j J_{ij}|$. Inset shows that average of the external input weights $\frac{1}{N} \sum_i w_i^{\text{in}}$ becomes more positive over epochs (green line) while the average row sum of the recurrent weights (black line) become more negative. **D)** The loss across learning epochs. **E)** The normalized dynamic gain $G(f)$ before and after training indicates improved high-frequency encoding as the network becomes more tightly balanced. **F** The mutual information between the time-varying external input and the linear readout as a function of training epochs. Information rate increases throughout training. Model parameters at initialization: $N = 100$, $g = \sqrt{2}$, $\Delta t = 0.01\tau$, $t_{\text{sim}} = 10\tau$, $b = 1$, $I_0 = J_0 = 1$, $I_1 = 0.1$, $\tau_S = 0.1$, $\sigma = 1$, epochs$= 10^5$.

while the average row sum of the recurrent weights (Figure 3C inset, black line) become more negative.

Consistent with our theory, we find that the mutual information rate (Gaussian channel approximations) grows over training epochs as the network becomes more tightly balanced (Fig 3F). Note that we did not train the network to maximize the mutual information rate, but to minimize the mean-squared error between target output and actual output, but the two are closely related [28]. We note that this tracking does not rely on the threshold-linear input-output function, in fact, we also successfully trained sigmoid (Fig. 9), linear (Fig. 10) and threshold-quadratic input-output function $\phi$ (Fig. 11) (see appendix H for additional results).

# 7 Training on multiple inputs results in tightly balanced subnetworks

We note that networks trained to simultaneously track multiple input Ornstein-Uhlenbeck signals by a linear readout exhibit a spontaneous symmetry breaking into weakly connected subnetworks with strong local inhibition (Fig. 4 and see appendix F for results on training on multiple signals). We note that this finding is not limited to training networks on two independent OU-processes; strongly inhibitory coupled subnetworks also emerge when training on more OU-processes (See Figure 4F for the resulting connectivity of a network trained on 3 OU-processes). For networks trained on multiple time-varying stimuli, we observed an interesting training dynamics. First, they develop a single global inhibitory mode (figure 7), similar to networks trained to track a single OU-signal. After some time, we observe the emergence of a second outlier eigenvalue. Concomitantly, the network breaks into two inhibitory subnetworks and the readouts start to track both OU-input signals independently.

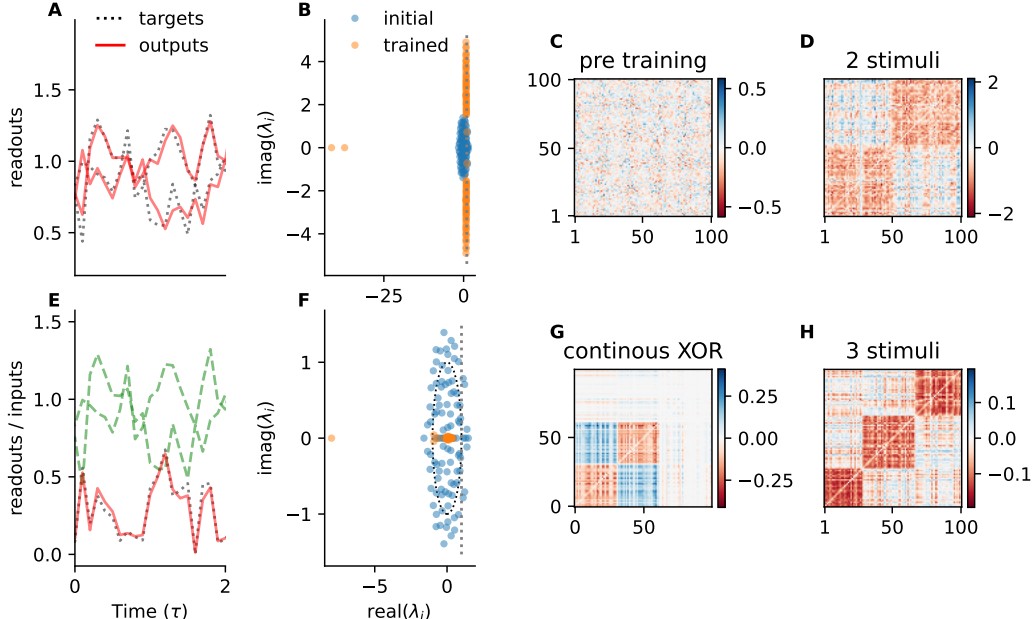

Figure 4: **Training on multiple signals and nonlinear task yields balanced subnetworks.**
**A)** RNNs are trained to approximate two time-varying external inputs $I_i(t)$ by two linear readouts $\hat{I}_i(t) = 1/N \sum_j w_{ij}^{\text{out}} \phi(h_j)$, by minimizing the mean squared loss $l = \sum_i \int |\hat{I}_i(t) - I_i(t)|^2 \, dt$.
**B)** Two eigenmodes with a strongly negative real part emerge. **C)** An example random recurrent network $J_{ij}$ depicted at initialization. Note that input and output weights are also random. **D)** Same example network after trained on two independent OU-processes. The network spontaneously breaks into two balanced subnetworks that have strong local inhibition but weak connectivity in between throughout training. Clustered by k-means clustering on the input weights. **E)** Same RNNs trained to calculate on the two time-varying external inputs $I_i(t)$ the continuous-time XOR (the function $f(I_1(t), I_2(t)) = |I_1(t) - I_2(t)|$ (Green dashed line is input, black dotted line is target output $f(I_1(t), I_2(t))$, red line is actual output). **F)** For XOR one eigenmode with a strongly negative real part emerges. **G)** Recurrent weight matrix after training on XOR shows balanced subnetworks that have strong local inhibition. Clustering by k-means clustering of input weights. **F)** An example recurrent network $J_{ij}$ that is trained to track 3 OU-processes develops three balanced subnetworks. Model parameters at initialization: $N = 100$, $g = \sqrt{2}$, $\Delta t = 0.2\tau$, $t_{\text{sim}} = 100\tau$, $b = 1$, $I_0 = J_0 = 1$, $\tau_S = 0.1$, $\sigma = 1$, epochs$= 10^5$.

# 8 Training on nontrivial computation results in tightly balanced network

Finally, we demonstrate the generality of our findings on two nontrivial tasks where the RNN not only copies a time-varying input but performs a computation on the input streams.

Firstly, we trained recurrent networks that receive two time-varying input signals to output a continuous extension of the XOR function $f(I_1(t), I_2(t)) = |I_1(t) - I_2(t)|$, which satisfies for $I_1, I_2 \in 0, 1$ the binary XOR function (Fig 4D). A similar, but not time-dependent task was used earlier to demonstrate that balanced networks can do nonlinear computations [29]. Consistent with our theory, we observe that the networks developed an outlier eigenvalue with strongly negative real part (Fig 4A). K-means clustering of the recurrent weight matrix revealed that the RNNs developed tightly balanced subnetworks throughout training (Fig 4E). See appendix F for further details. Note that a substantial fraction of neurons in the RNN are silent after training, we leave a further analysis of these silent fraction for future work.

Secondly, we trained RNNs to compute a linear transformation of a time-varying vector of stochastic processes $y_i(t) = \sum_j A_{ij} I_j(t)$ at every moment in time. We observed that the RNNs develop a number of negative outlier eigenvalues and concomitantly the same number of tightly balanced inhibitory subnetworks that corresponds to the rank of the linear input-output transformation $A$ (See Fig 12 and appendix F for further details).

# 9 Limitations

Here we considered the encoding of multivariate scalar input signals only numerically, it remains an important challenge to extend our theory to multidimensional input signals, including the training dynamics. For mathematical tractability, we considered firing rate networks here, but for more detailed neuron models, other biophysical features shape the information encoding rate. For example, in spiking neuron models, fluctuating background input can enhance the high-frequency encoding [30]. Moreover, the spike generation mechanism [31, 32, 33, 34, 35, 36], the synaptic dynamics ([37, 38]) and other latent variables (e.g. adaptation [39]) shape the frequency-response. It remains an important future work to extend the work here by such contributions. Moreover, moving beyond point-neurons, the shape of the dendritic tree can also affect the frequency response [36].

While we show that the dynamics of RNNs trained on several linear and nonlinear tasks is consistent with our DMFT, it remains an important challenge to consider tasks with more complex temporal structure, e.g. involving memory, where the task dynamics might interfere with the tightness of balance.

Finally, our theory becomes exact only in the limit of large network size, and it is important to also consider finite-size effects. For finite network size, both noise and chaotic fluctuations also contribute to fluctuations of the mean $m$, which are also recurrently canceled. This was described previously in recurrent networks in a linear regime [27]. we merely provide a heuristic extension of the DMFT for finite-size fluctuations (Appendix B), but a full theory would be desirable.

# 10 Discussion

We show how a tight balance of excitatory currents by recurrent inhibition improves information encoding of a time-varying signal. We demonstrate that the mutual information rate of a time-varying signal increases linearly with the tightness of balance, both in the presence of additive noise and with chaotic fluctuations of the recurrent network activity. A non-stationary dynamic mean-field theory reveals a separation of timescale between the mean currents and the timescale of residual fluctuations. The mean dynamics become linearly faster with the tightness of balanced and enable reliable encoding of time-varying signals. In contrast, the timescale of the chaotic dynamics in the residual fluctuations are largely unaffected by the tightness of balance. We find that networks become more robust to deteriorating effects of fluctuations from noise and chaos as the network becomes more tightly balanced.

Our study is relevant in the recent debate on the functional implications of how tightly excitatory currents are tracked by recurrent inhibition [26, 27, 7, 10, 13, 14]. We address this question by building a bridge from information-theoretic measures of information encoding that were previously used in neuroscience mostly in sensory systems [40, 41, 42, 23, 24] to dynamic mean-field descriptions of recurrent network dynamics that were previously used to describe the often chaotic dynamics of recurrent rate networks [18, 19, 20, 16, 43, 17, 44, 14].

Besides the implications on information encoding of more loosely or more tightly balanced networks, of course also biophysical, energetic, and evolutionary constraints should be considered. Biological networks can naturally not be arbitrarily tightly balanced, which would require arbitrary large synaptic currents. Very tight balance might also be questionable for energetic reasons, as was asked previously, "Why should the cortex simultaneously push on the accelerator and on the brake?" [45]. However, such biophysical and energetic constraints may be better addressed in biophysically more detailed models.

We found that training networks on tracking a time-dependent signal by a linear readout by minimizing the squared error make them more tightly balanced, as reflected in more negative mean recurrent weights and more positive feedforward input weights. Moreover, the training also increased the mutual information rate between the input signal and linear readout. This is consistent with our theoretical result. Furthermore, the fact that training arrives at a tightly balanced solution may suggest that this is a typical solution for a network. It is an interesting question when such balanced solutions will emerge with biologically plausible learning rules [46, 47, 48, 49])

Previous studies on the effect of time-varying input in chaotic rate networks were limited to independent inputs across neurons in the form of stochastic [15, 17] and sinusoidal [16] drive, but the networks were not balanced, and their connectivity had zero mean coupling. In these previous studies, the distribution of inputs across the population is time-independent [15, 16, 17] and stationary dynamic mean-field theory was sufficient to describe the results. However, the treatment of common input is only possible by the non-stationary dynamic mean-field approach introduced here.

The dynamic cancellation of time-varying input through recurrent inhibitory feedback has been previously studied in balanced networks with binary [2, 26] and spiking neurons [27, 50, 51]. Chaos in balanced firing-rate networks was studied previously [19, 20, 52, 17], but the dynamic cancellation of correlated input and its implications on information encoding in rate networks were not investigated, nor were the implications for training such networks in a machine learning setup with backpropagation through time. It would be interesting to extend our mean-field analysis to rate networks with pre-existing low-rank structures on top of the random structure [53, 54, 55, 56].

The underlying mechanisms of tracking of time-varying input we analyze here are not specific to fully-connected threshold-linear RNNs driven by Ornstein-Uhlenbeck signals with additive Gaussian white noise, which we merely chose for the sake of simplicity and analytical tractability. Rate networks with other non-negative input-output transfer functions exhibit a qualitatively similar dynamic tracking in the tightly balanced regime. Moreover, the mechanism of tracking described here is closely related to balanced predictive coding networks [13, 14].

A testable hypothesis based on our findings is that changing the tightness of balance, e.g. by pharmacological manipulation or by a genetic knockout that affects net recurrent inhibitory strength (without generating runaway activity or pathological network states), would also affect the high-frequency stimulus encoding and information transmission. Specifically, we predict that stronger effective recurrent weights in conjunction with stronger excitatory inputs would improve both the high-frequency encoding and the mutual information rate between a stimulus and the population response. Conversely, when weakening recurrent inhibition and external input strength, we predict an impaired high-frequency encoding and a lower mutual information rate. Such predictions could not only be tested in vivo, but also through in vitro experiments, where the number of recurrent synapses can be manipulated [57].

## Acknowledgments and Disclosure of Funding

Research supported by NSF NeuroNex Award DBI-1707398 (RE), the Gatsby Charitable Foundation (RE), the Bernstein Award 2014, 01GQ171 (SG), and the Swartz Foundation Award (2021-6) (RE). We thank L.F. Abbott, A. Ingrosso, C. Cueva, J. Kadmon, J. Keijser, R. Khajeh, L. Logiaco, A. Neef, K. Rajan, C. Machens, K. Miller, T. Nguyen, S. Solla, C. v. Vreeswijk (†) and F. Wolf for fruitful discussions. The funders had no role in study design, data collection and analysis, decision to publish, or preparation of the manuscript.

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
