# A  Non-stationary dynamic mean-field theory with both independent and common input

The core idea of dynamic mean-field theory is that for large $N$, the distribution of the residual recurrent input for different neurons becomes Gaussian and pairwise uncorrelated, according to the central limit theorem. To this end, we characterize the distribution of the $\tilde{h}_i(t)$ (Eq 3b) by considering the (linear) stochastic dynamics:

$$\tau \frac{\mathrm{d}\tilde{h}(t)}{\mathrm{d}t} = -\tilde{h}(t) + \eta(t) + \xi(t)\,, \tag{9}$$

where $\eta(t)$ is a Gaussian process with mean $\langle \eta(t) \rangle = 0$ and autocorrelation

$$q(t,s) = \langle \eta(t)\eta(s) \rangle = g^2 \langle \phi(m(t) + \tilde{h}(t))\phi(m(s) + \tilde{h}(s)) \rangle\,. \tag{10}$$

Here and in the following, angular brackets denote expectation values over the distribution of the stochastic process $\tilde{h}(t)$, which approximates population averages in the full network. The mean-field estimate for the mean $m(t)$ therefore evolves according to Eq 3a with $\nu(t) = \langle \phi(m(t) + \tilde{h}(t)) \rangle$, the mean-field estimate of the mean population firing rate.

We obtain an expression for the time evolution of the two-time autocorrelation function $c(t,s) = \left\langle \tilde{h}(t)\tilde{h}(s) \right\rangle$, which explicitly depends on two time points. Taking the temporal derivative of $c(t,s)$ with respect to $s$ and using Eq 9, we obtain

$$\tau \frac{\mathrm{d}}{\mathrm{d}s}c(t,s) = -c(t,s) + r(t,s)\,, \tag{11}$$

where $r(t,s) = \left\langle \tilde{h}(t)\left(\eta(s) + \xi(s)\right) \right\rangle$, which we take as an auxiliary function. Taking the temporal derivative of $r(t,s)$ with respect to $t$ and using Eq 9 again, we arrive at

$$\tau \frac{\mathrm{d}}{\mathrm{d}t}r(t,s) = -r(t,s) + q(t,s) + \tau\sigma^2\delta(t-s)\,, \tag{12}$$

where $q(t,s) = g^2 \langle \phi(m(t) + \tilde{h}(t))\phi(m(s) + \tilde{h}(s)) \rangle$ (see Eq 10) and $\sigma^2$ is the variance of fluctuations driven by the independent noise input. The additional term proportional to $\delta(t-s)$ in Eq. 12 is due to this independent additive Gaussian white noise, similarly to the independent input considered previously [17, 16]. However, we consider simultaneously a time-varying common input signal $I(t)$, which renders the problem non-stationary. Such a non-stationary DMFT was recently developed [21], but without the independent input, which is crucial for meaningfully discussing information encoding rates in terms of a signal-to-noise ratio.

Together, the dynamic mean-field equations for $m(t)$, $c(t,s)$ and $r(t,s)$ form a closed system of self-consistent dynamic equations and can be solved forward in time $s$ and $t$ by integrating them on a two-dimensional grid from some initial condition for $m$, $c$ and $r$. The integration requires $q(t,s)$, which can be calculated by evaluating a Gaussian double integral that depends on $c(t,s)$, $c(t,t)$, $c(s,s)$, $m(t)$ and $m(s)$. For the threshold-linear transfer function $\phi(x) = \max(x,0)$, one integral can be evaluated analytically, which allows for an efficient numerical implementation using adaptive Gauss–Kronrod integration. The non-stationary dynamic mean-field theory captures accurately the time-dependent mean population rate $\nu(t)$ and the two-time autocorrelation function from numerical simulations.

In contrast to previous work that did not include noise [21], we consider here a network where neurons simultaneously receive a time-varying input component $I(t)$ ("signal") that is identical across neurons and an input component $\xi_i$ ("noise") that is independent across neurons. Furthermore, we go beyond our previously published non-stationary dynamic mean-field theory [21] by introducing a heuristic term that emulates fluctuating input into the equation of the mean $m$ due to finite network size $N$ similar to [14] (see appendix B). We note that in contrast to [14], we do solve the self-consistent set of dynamic mean-field equations for $m$, $c$ and $r$ to calculate dynamic gain, mutual information rate and mutual information rate density.

# B  Heuristic finite-size fluctuations

The non-stationary dynamic mean-field theory is exact in the limit of large $N$. However, for the precise evaluation of the spectral coherence, finite-size network corrections have to be taken into account. These corrections are expected to be of order $\frac{1}{\sqrt{N}}$ and therefore have only a marginal effect on the dynamic gain. The finite-size fluctuations can be heuristically accounted for by adding a fluctuation term to the equation of $m(t)$ (mean-field version of Eq. 3a), analogously to previous work [14]:

$$\tau \frac{\mathrm{d}m}{\mathrm{d}t} = -m - bJ_0\nu(t) + bI(t) + \frac{1}{\sqrt{N}}\zeta(t)\,, \tag{13}$$

where the mean-field estimate of the mean population firing rate $\nu(t) = \langle \phi(m(t) + \tilde{h}(t)) \rangle$ is additionally driven by the fluctuations $\zeta(t)$, a noise term that captures the effective finite-size fluctuations. In the case of $g = 0$, $\zeta(t)$ reduces to an additive white Gaussian noise processes (AWGN) with autocorrelation function $\langle \zeta(t)\zeta(t+t') \rangle = \tau\sigma^2\delta(t')$ originating from the mean across neurons of the independent input noise $\xi_i(t)$.

For $g > 0$, the finite size noise $\zeta(t)$ should include the effect coming from the residual fluctuations $\tilde{h}_i(t)$, which adds a colored noise component to it (see also [14]). In contrast to [14], we heuristically approximate both amplitude and timescale of the finite-size fluctuations from the solution of the stationary DMFT.

## C   Details on numerical and analytical spectral analysis

We numerically solve the stochastic differential equation (Eq. 1) using the Euler–Maruyama method but with a distributionally-exact solution of the Ornstein-Uhlenbeck process for $I(t)$, meaning that $I(t)$ is not a numerical solution of stochastic differential equations, but obtained through exact sampling of the (continuous-time) solution of the SDE. To calculate the spectral coherence, we used the function *coherence* from the Python package *scipy.signal* version 1.7.3 with a default Hann window and default parameters except that we set the length of each segment (nperseg) to $2^{11}$. We calculated the integral over the mutual information rate density $1 - C_{I\nu}(f)$ using standard trapezoidal integration. As pointed out in the main text, we find that the mutual information rate approximately grows linear with tightness of balance $b$. In figure 5, we demonstrate that this linear scaling over several orders of magnitude is also true for different levels of additive Gaussian white noise $\sigma$. We note that for low noise $\sigma$ and large $b$, the mutual information rate saturates which is an effect of finite time-resolution of the integration of both dynamic mean-field theory and direct numerical simulations that result in an effective upper bound on the frequencies (see figure 5), which leads to a deviation from the linear scaling of the mutual information rate. We confirm this by deliberately introducing an upper cutoff frequency $f_{\text{cutoff}}$ up to which the spectral coherence is integrated:

$$R_{\text{lb}}(I, \nu) = -\int_0^{f_{\text{cutoff}}} df \log_2\left(1 - C_{I\nu}(f)\right) \tag{14}$$

As expected, changing this frequency cutoff directly affects the mutual information rate for large $b$ (see figure 6). For high noise $\sigma$ and small $b$, we also observe deviation from the linear scaling. In that case, for finite simulation time $t_{\text{sim}}$, the signal is so weak that spurious correlations between input and output give the illusion of mutual information. This lower saturation can be ameliorated by a longer simulation time. All simulations were performed on a laptop and took minutes to hours.

## D   Details on analytical approximation of mutual information rate

We observed an approximately linear scaling of the mutual information rate with tightness of balance $b$, both for noise-driven homogeneous networks ($g = 0$) and heterogeneous networks ($g > \sqrt{2}$) that receive an OU-process as input. Where does this linear scaling originate from? The linear scaling of the mutual information rate can be obtained from the boost of the effective timescale of the network in combination with the signal-to-noise ratio that comes from the OU-input signal and the additive Gaussian white noise. The boost of the effective timescale was already noted earlier [2, 14], but we provided a non-stationary mean-field theory for nonlinear rate networks and measured the effect on the mutual information rate. The power spectral density of the input signal is given by

$$S_{\text{signal}}(f) = \begin{cases} \frac{2\tau_S I_1^2}{1 + (2\pi\tau_S f)^2} & f \leq f_{\text{cutoff}} \\ 0 & \text{else.} \end{cases} \tag{15}$$

$S_{\text{noise}}(f)$ is the power spectrum of band-limited Gaussian white noise:

$$S_{\text{noise}}(f) = \begin{cases} 2\sigma^2\tau & f \leq f_{\text{cutoff}} \\ 0 & \text{else.} \end{cases} \tag{16}$$

In an approximation that ignores the nonlinearity of the recurrent network dynamics and the interplay between $m$ and $\tilde{h}$, we can approximate that $b$ changes the effective low-pass filtering of the network from the low-pass filtering coming from the leak-term in the dynamics

$$G_{\text{leak}}(f) = 1/\sqrt{1 + 4\pi^2 f^2 \tau^2} \tag{17}$$

to an effective filter of

$$G_{\text{effective}}(f) = 1/\sqrt{1 + 4\pi^2 f^2 \tau^2 / b^2}. \tag{18}$$

Consequently, the characteristic timescales of the mean $m(t)$ and therefore of the population firing rate $\nu(t)$ is $\tau/b$. This results from Eq. 3, when assuming a separation of timescales between the time-varying statistics of $m$

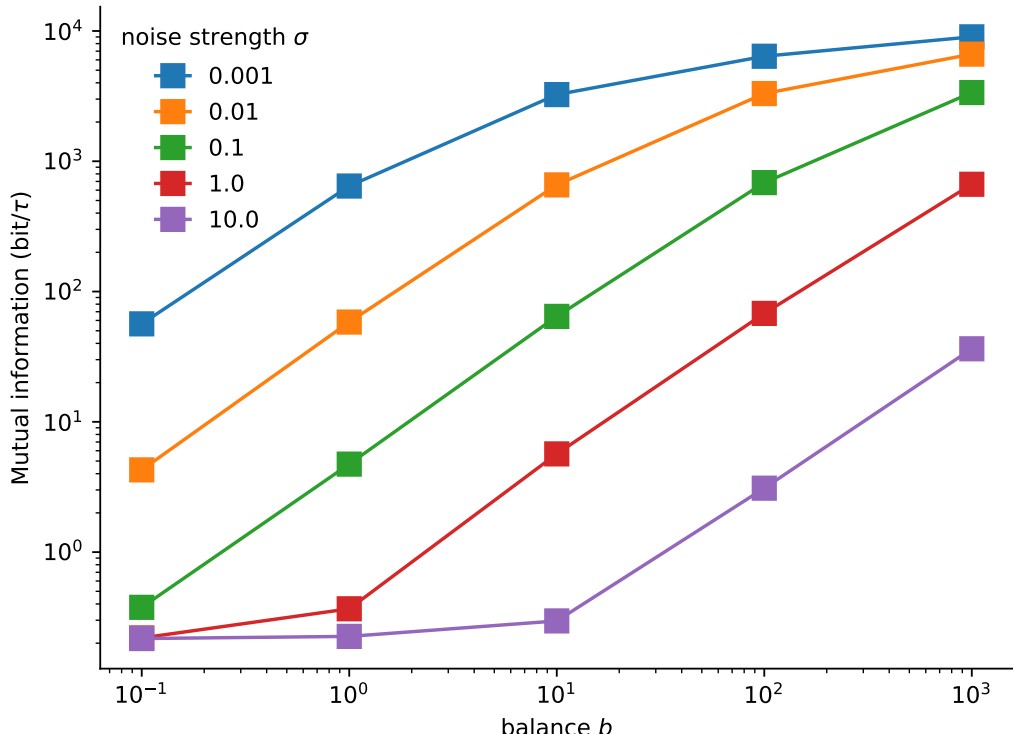

Figure 5: **Information rate grows linearly with b for different noise strength $\sigma$.** Same as figure 1E but with different additive white Gaussian noise of strength $\sigma$. Model parameters: $N = 64$, $g = 0$, $\Delta t = 1/2^{12}\tau$, $t_{\text{sim}} = 2^6\tau$, $I_0 = J_0 = 1$, $\tau_S = \tau$, $\sigma = 0.1$, $I_1 = 1/8$.
f

and $\tilde{h}$. Thus, in a linear approximation, the magnitude square coherence can be approximated to be

$$C_{I\nu}(f) = \frac{|S_{I\nu}(f)|^2}{|S_{II}(f)||S_{\nu\nu}(f)|} \approx \frac{|S_{\text{signal}}(f)|}{|S_{\text{signal}}(f)| + |S_{\text{noise}}(f)|/(Nb^2)}. \tag{19}$$

Using the equation for the mutual information rate in the Gaussian channel approximation, this results in a mutual information rate density of

$$r_{\text{lb}}(f) \approx -\log_2\left(1 - \frac{|S_{\text{signal}}(f)|}{|S_{\text{signal}}(f)| + |S_{\text{noise}}(f)|/(Nb^2)}\right) = \log_2\left(1 + Nb^2\frac{|S_{\text{signal}}(f)|}{|S_{\text{noise}}(f)|}\right). \tag{20}$$

Thus, the effective signal-to-noise ratio of a scalar signal encoded in the population rate is boosted by a factor $Nb^2$. Calculating the mutual information rate gives

$$R_{\text{lb}} = \int_0^\infty df \log_2\left(1 + \frac{Nb^2}{\sigma^2\tau}\frac{\tau_S I_1^2}{1 + (2\pi\tau_S f)^2}\right) = \frac{\left(\sqrt{\frac{N\tau_S I_1^2 b^2}{\sigma^2\tau} + 1} - 1\right)}{2\tau_S \log 2} \tag{21}$$

for $bI_1\sqrt{\tau_S/\tau} \gg \sigma/\sqrt{N}$, i.e., for large networks and large signal-to-noise ratio, this results in a linear scaling of the mutual information rate with tightness of balance:

$$\lim_{bI_1\sqrt{\tau_S/\tau} \gg \sigma/\sqrt{N}} R_{\text{lb}} = \frac{bI_1\sqrt{N}}{2\sigma\sqrt{\tau_S/\tau}\log 2}. \tag{22}$$

We observe already for moderate tightness of balance and moderate network size very good agreement between this linear approximation and numerical experiments.

## E   Details on training setup and additional numerical training experiments

An example implementation in Flux, a machine learning library in Julia is available at training-balanced-nets. Briefly, we trained standard rate networks ( Eq. 1) to track an OU-process by a linear readout. We implemented

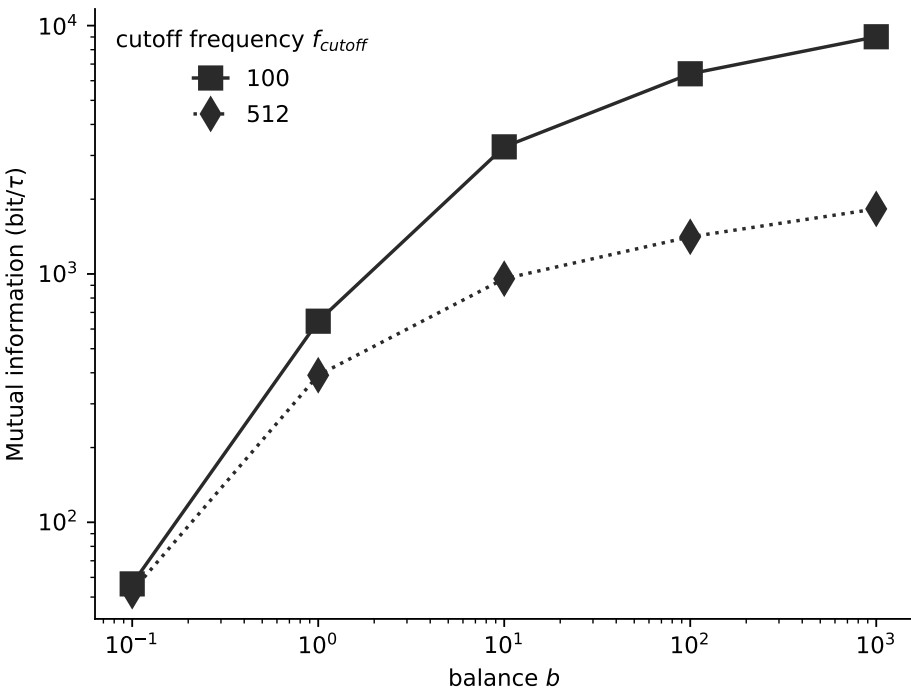

Figure 6: **Cutoff frequency of input signal** $I(t)$ **limits linear scaling of mutual information.** Same as figure1E but with different cutoff frequency $f_{\text{cutoff}}$. Model parameters: $N = 64$, $g = 0$, $\Delta t = 1/2^{12}\tau$, $t_{\text{sim}} = 2^6\tau$, $I_0 = J_0 = 1$, $\tau_S = \tau$, $\sigma = 0.001$, $\sigma = 0.1$, $I_1 = 1/8$.

the stochastic dynamics using the Euler–Maruyama method. We used the default ADAM hyperparameters without any additional fine-tuning and found our results to be robust with respect to choices of parameters. The parameters of the network simulations are reported in the respective figure captions.

## F    Details on training networks to track multiple time-varying signals

We found that training recurrent neural networks simultaneously on multiple signals yields balanced subnetworks. We investigated this in the following scenario: We trained RNNs to approximate two time-varying external inputs $I_i(t)$ by two linear readouts $\hat{I}_i(t) = 1/N \sum_j w_{ij}^{\text{out}} \phi(h_j)$, by minimizing the mean squared loss $l = \sum_i \int |\hat{I}_i(t) - I_i(t)|^2 \, dt$. The training scenario is identical to training on tracking a single OU-process, the only difference is that there exist two independent OU-input processes $I_i(t)$ and two readout vectors $w_i^{\text{out}}$. All parameters (recurrent weight matrix $J_{ij}$, input weights and output weights) are initialized with parameters drawn independently identically from Gaussian distributions as before.

We found that such randomly initialized networks trained on two OU-processes spontaneously break into two balanced subnetworks throughout training. The subnetworks have strong local inhibition among neurons belonging to one subnetwork, but weak connectivity between the subnetworks. We displayed that finding in Figure 4A-D. We find the two subnetworks by k-means clustering on the input weights. Note that in the case of two OU-processes, the sorting can also be obtained by simply sorting the weight matrix $J_{ij}$ according to the indices of the sorted input weight vectors, i.e. by performing 'sortperm' on one of the input vectors using the resulting indices as row- and column indices of the displayed weight matrix $J_{ij}$. Furthermore, we found that the entries of the input weights, which were independent Gaussian random initially also displayed symmetry breaking. We note that this finding is not limited to training networks on two independent OU-processes; strongly inhibitory coupled subnetworks also emerge when training on a larger number of OU-processes (See Figure 4F for the resulting connectivity of a network trained on three OU-processes). For networks trained on multiple time-varying stimuli, we observed an interesting training dynamics. First, they develop a single global inhibitory mode (figure 7), similar to networks trained to track a single OU-signal. After some intermediate time, we observe the emergence of a second negative outlier eigenvalue. Concomitantly, the network breaks into

two inhibitory subnetworks and the readouts start to track both OU-input signals independently. An analytical analysis of this phenomenon is beyond this publication, but it can be explained in terms of the network to learn subsequent singular values of the input-output cross-correlation matrix [58], where the singular value that captures most of the variance is learned first. Indeed, tracking the average of the two OU-signals at each moment in time is favorable compared to e.g. tracking only one and ignoring the other. This can be seen by calculating the expectation of the mean squared error for the two cases. In the case where the mean is perfectly tracked, the mean squared error is by a factor of two larger in case only one is tracked and the other is ignored.

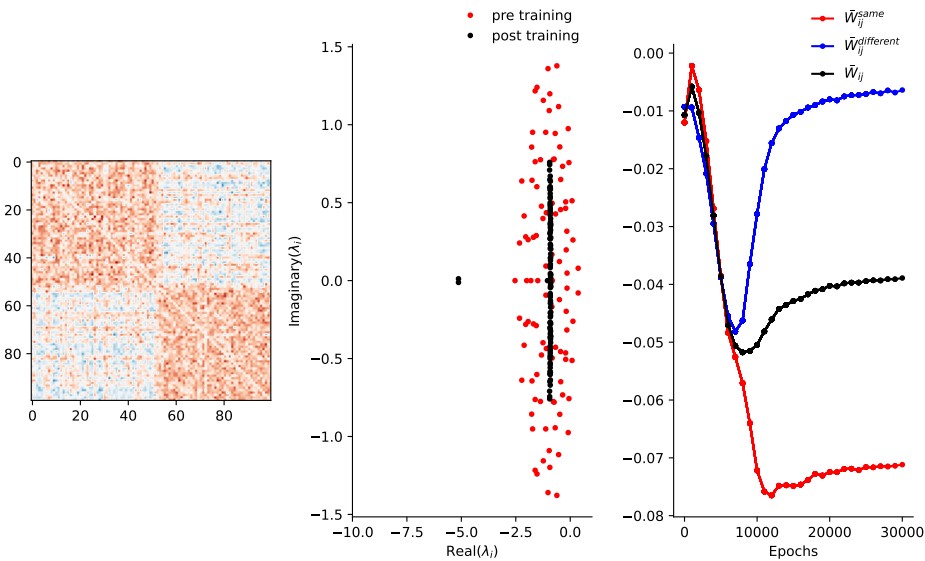

Figure 7: **Training dynamics of networks trained on multiple signals shows first tracking of global mean input. A)** RNNs are trained to approximate two time-varying external inputs $I_i(t)$ by two linear readout $\hat{I}_i(t) = 1/N \sum_j w_{ij}^{\text{out}} \phi(h_j)$, by minimizing the mean squared loss $l = \sum_i \int |\hat{I}_i(t) - I_i(t)|^2 \, \mathrm{d}t$. **B)** When trained on two independent stimuli, two outlier eigenvalues with negative real parts emerge. **C)** When observing the mean connectivity strength within the two subnetworks and between and the global mean coupling, an interesting training dynamics emerges: First, all weights become more negative and one single negative outlier eigenvalue emerges, then at some intermediate state, the network breaks up into two subnetworks, and their respective means become more negative but the weights between them on average actually become closer to zero. Model parameters at initialization: $N = 100$, $g = \sqrt{2}$, $\Delta t = 0.2\tau$, $t_{\text{sim}} = \tau$, $b = 1$, $I_0 = J_0 = 1$, $\tau_S = 0.1$, $\sigma = 1$, epochs$= 30000$.

## G   Mixed excitatory-inhibitory circuits

To corroborate our training results, we also initialized the trained network with separate excitatory and inhibitory populations using the classical parametrization of [1], which for large $b$ results in a balanced state. Consistent with [1], we chose mean couplings $J_{EE} = J_{IE} = 1.0/N$, $J_{II} = -1.8/N$, $J_{EI} = -2.0/N$. We found that training such a network on the auto-encoder task also results in a single inhibition-dominated population (See Figure 7). We did not try to constrain the weight update to maintain a separation of excitation and inhibition. This interesting scenario is left for future work.

## H   Training networks with other nonlinearities

To corroborate our training results and show the generality of our findings, we trained networks on the autoencoding task also using sigmoid (Figure 9), linear (Figure 10) and threshold-quadratic (Figure 11) activation function $\phi(x)$. We found that training such networks on the auto-encoder task also results in a single inhibition-dominated population (See Figure 9 - 11) that becomes progressively more tightly balanced over time, as seen from the outlier eigenvalue that emerges over time.

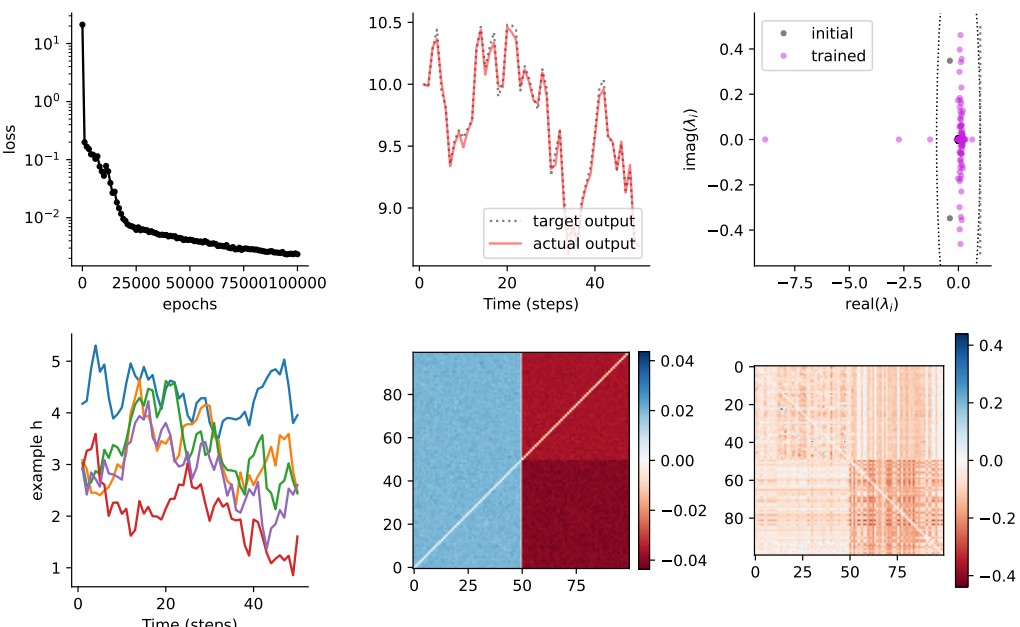

Figure 8: **Excitatory-inhibitory initialization lost during training.** The network is initialized with excitatory and inhibitory subnetworks using the classical parametrization of [1]. **Top right:** Decay of mean squared error over epochs indicates that the network trained successfully. **Top middle:** This is confirmed visually by testing it on tracking an OU-process it was not trained on (target output dotted, actual output red). **Top right:** During training, an outlier eigenvalue with a very negative real part emerges similar to Fig.3. **Bottom left:** Typical activity of five example neurons after training. **Bottom middle:** An example recurrent connectivity matrix is displayed before training. **Bottom right:** After 50000 epochs, the initialization with separate excitatory and inhibitory subnetworks is largely lost and the network learned to track the time-varying external input. Model parameters at initialization: $N = 100$, $g = 0.01$, $\Delta t = 0.2\tau$, $t_{\text{sim}} = 100\tau$, $b = 1$, $I_0 = J_0 = 1$, $\tau_S = 0.1$, $\sigma = 1$, epochs$= 10^5$. Mean couplings were $J_{EE} = J_{IE} = 1.0/N$, $J_{II} = -1.8/N$, $J_{EI} = -2.0/N$.

## I Training on linear transformation on multiple input streams results in tightly balanced subnetworks

To further corroborate our training results and show the generality of our findings, we also trained networks on calculating a linear transformation on multiple input streams in each moment in time.

For that, analogous to Fig.4B-D, RNNs are trained to perform a linear transformation $y_i(t) = \sum_j A_{ij} I_j(t)$ on a set of multi-dimensional time-varying external inputs $I_i(t)$ (Fig 12 top middle). The training was done by minimizing the mean squared loss $l = \sum_j \int |\hat{y}(t) - \sum_{ij} A_{ij} I_j(t)|^2 \, \mathrm{d}t$, where $\hat{y}_i$ are linear readouts of the recurrent network, $\hat{y}_i(t) = 1/N \sum_j w_{ij}^{\text{out}} \phi(h_j)$ and $A$ is a random matrix of size $S \times R$ with entries drawn independently from a Gaussian distribution with zero mean and variance $1/R$. Here we chose $R = S = 3$. Again, the time-varying input signals $I_j(t)$ were independent OU processes.

We observe that the test loss drops throughout training (Fig 12 top left), concomitantly the linearized network dynamics develops three outlier with negative real part throughout the course of the training (Fig 12 top right). We observe that the number of outlier eigenvalues is determined by the rank of $A$ (Fig 12 bottom left) and their training dynamics seems to be shaped by the singular values of $A$, reminiscent of the training dynamics of multi-layer perceptrons [58]. For our case of $R = S = 3$ in figure 12, we observed that three outlier eigenvalues develop, at the same time three tightly balanced subnetworks with strong inhibition emerge (Fig 12 bottom right). For illustration, we plotted the activity $h_i$ for 5 example neurons in the bottom middle of Fig 12. Consistent with our theory, the network becomes more tightly balanced over the course of training. We leave an analytical description of the training dynamics for future work.

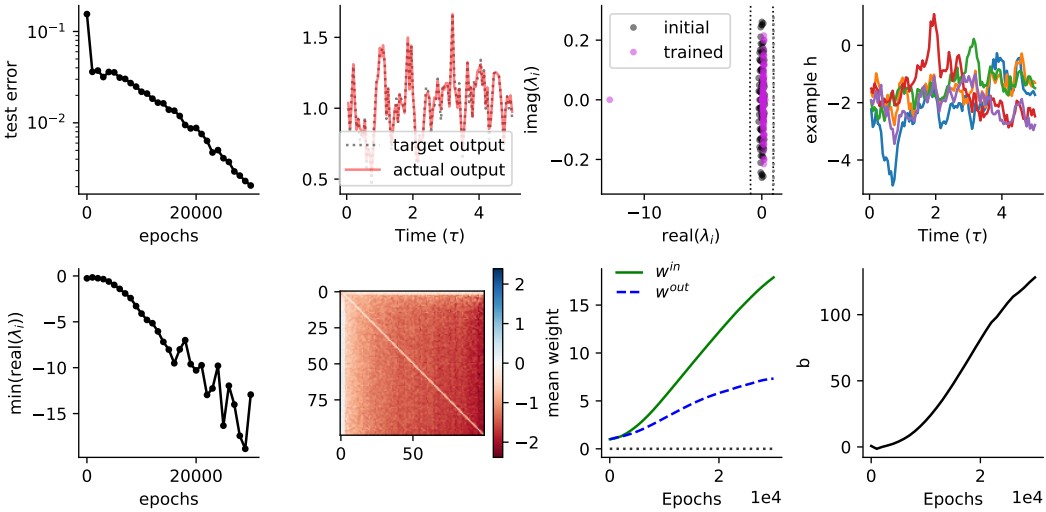

Figure 9: **RNNs with sigmoid nonlinearity trained on tracking time-varying input become more tightly balanced throughout training. Top left)** Analogous to Fig.3, RNNs are trained to approximate a time-varying external input $I(t)$ by linear readout $\hat{I}(t) = 1/N \sum_i w_i^{\text{out}} \phi(h_i)$, by minimizing the mean squared loss $l = \int |\hat{I}(t) - I(t)|^2 \, \mathrm{d}t$. Test loss drops throughout training. **Top middle left)** Target output $I(t)$ (black dashed) and actual linear readout $\hat{I}(t)$ are plotted after training. **Top middle right)** The eigenvalue spectrum of the dynamics linearized at the fixed point before and after training. **Top right)** Typical activity $h_i(t)$ of five example neurons after training. Note that individual neurons have strong slow fluctuations despite reliable network response. **Bottom left)** The training dynamics of the minimum over the real parts of the eigenvalues indicates that during training, an eigenmode with a strongly negative real part emerges. **Bottom middle left)** Example network after training on independent OU-processes. After training, the network has an overall strongly negative recurrent weight matrix, despite being initialized with weights drawn from a Gaussian with a mean $-bJ_0/N = -0.01$. **Bottom middle right)** Mean external input weights $\frac{1}{N} \sum_i w_i^{\text{in}}$ are increasing strongly during training (green line), consistent with theory, while output weights $\frac{1}{N} \sum_i w_i^{\text{out}}$ are only moderately increasing (dashed blue line). **Bottom right)** Consistent with theory, the network becomes more tightly balanced as the increasing empirical balance $b$ shows. Model parameters at initialization: $N = 100$, $g = \sqrt{2}$, $\Delta t = 0.05\tau$, $t_{\text{sim}} = 5\tau$, $b = 1$, $I_0 = J_0 = 1$, $\tau_S = 0.1$, $\sigma = 1$, epochs$= 30000$.

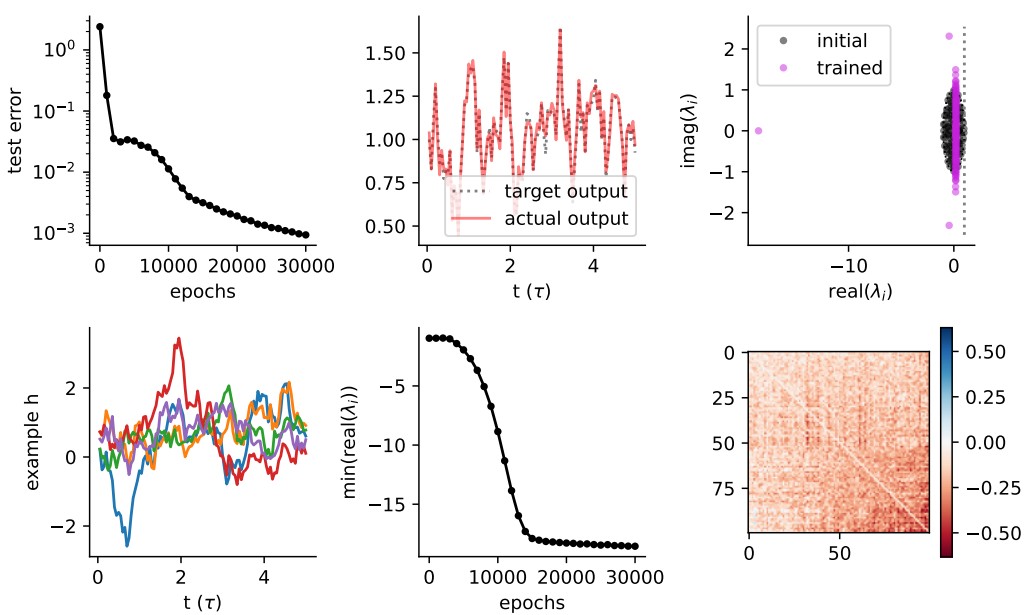

Figure 10: **Linear RNN trained on tracking time-varying input become more tightly balanced throughout training.** Everything same as Fig 9 but linear activation $\phi(x) = x$.

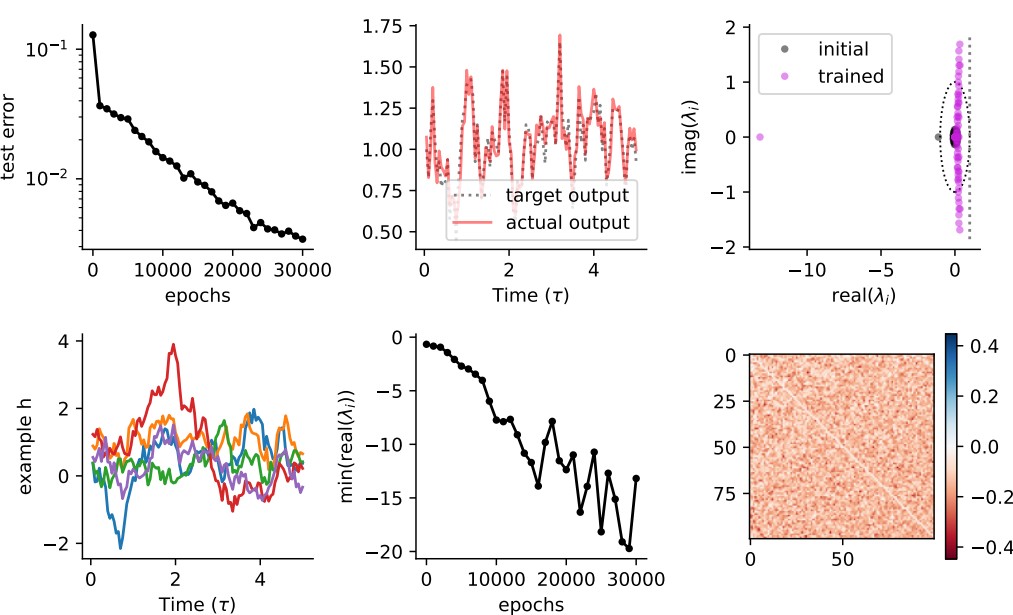

Figure 11: **RNNs with threshold quadratic nonlinearity trained on tracking time-varying input become more tightly balanced throughout training.** Everything same as Fig 9 but threshold quadratic activation $\phi(x) = max(x, 0)^2$.

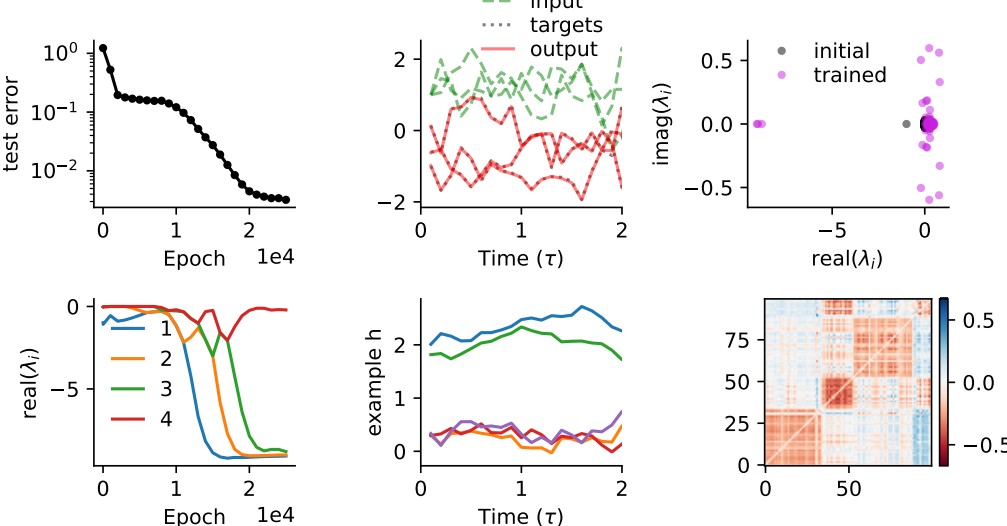

Figure 12: **RNNs trained on performing linear transformation on time-varying input become more tightly balanced throughout training. Top left)** Analogous to Fig.4B-D, RNNs are trained to perform a linear transformation $y_i(t) = \sum_j A_{ij} I_j(t)$ on a set of multi-dimensional time-varying external input $I_i(t)$, by minimizing the mean squared loss $l = \sum_i \int |\hat{y}_i - \sum_j A_{ij} I_j|^2 \, \mathrm{d}t$, where $\hat{y}_i$ is again a linear readout approximating $y_i$. Test loss drops throughout training. **Top middle)** Target output $y(t)$ (black dashed) and actual linear readout $\hat{y}(t)$ are plotted after training. **Top right)** The eigenvalue spectrum of the recurrent weights before and after training. After training, three negative outlier eigenvalues emerged. **Bottom left)** Training dynamics of the four most negative outlier eigenvalues. In this case (rank of $A_{ij}$ is 3), there are three outlier eigenvalues forming over time (c.f. [58]). **Bottom middle)** Typical activity $h_i(t)$ of five example neurons after training. **Bottom right)** Example recurrent weights after training on linear transformation of three OU-processes. After training, the network developed three strongly inhibitory subnetworks. Model parameters at initialization: $N = 100$, $g = 0.1$, $\Delta t = 0.05\tau$, $t_{\mathrm{sim}} = 5\tau$, $b = 1$, $I_0 = J_0 = 1$, $\tau_S = 0.1$, $\sigma = 1$, epochs= 30000.