# OpenReview forum: "A time-resolved theory of information encoding in recurrent neural networks"
_NeurIPS.cc/2022/Conference — NeurIPS 2022 Accept_

### Official Review · Reviewer_H6Fq · 2022-07-06

**Rating:** 7
**Confidence:** 4
**Soundness:** 2 fair
**Presentation:** 4 excellent
**Contribution:** 3 good

**Summary:**

This paper proposes a non-stationary dynamic mean-field theory to relate the tightness of the balance of a network to the quality of information encoding. More particularly, it shows that a tighter balance improves the mutual information rate. The theoretical analysis is backed up with numerical experiments. Finally, the paper shows that learning the input weights, output weights and recurrent weights of a network whose goal is to auto-encode its one-dimensional input increases the mean recurrent coupling of the network. A generalization to multi-dimensional inputs shows that the network breaks into subnetworks with intense local inhibitory connections and weak between-clusters connections.

**Questions:**

1. Could the context of "depth of balance" / "empirical balance" be used in other studies?
2. What is the difference between a binary network and a spiking network?
3. I suggest moving Fig 6 to the main text: it is a shame that you talk about it only in the abstract and in the supplementary material...
4. It would be interesting to consider unsupervised learning rules to learn the input/output/recurrent weights, instead of supervised learning
5. Section 8: Couldn't you have checked the finite-size effects through simulations? Or did you instead mean that the finite-size effects should be inspected analytically?
6. Provide a supplementary figure for the claims in the last paragraph of the discussion

----------------------------------
Minor issues (e.g. typos):
1. Eq 5: "m" should write "m/b" on the right-hand side
2. End of section 3: "in contrast to previous work [21]": precise that [21] has no noise
3. Problem in the definition of the dynamic gain G(f)
4. End of section 5: "transmitted more reliable" --> "reliably"
5. First sentence of section 6: formulation is wrong after "we analyse"
6. Beginning of section 7: rather use present tense ("corroborate")
7. Use "i.e." instead of "e.i." (appearing twice in the text)
8. Missing parentheses in the paragraph referring to Fig 3a/b/c
9. First paragraph of section 8: missing space before "the synaptic dynamics"
10. Appendix G: "we trained RNN are trained" (mistake)
11. Appendix G: definition of the mean-squared loss: missing sum on index i? Shouldn't it be \hat{x}_i rather than \hat{x}?
12. Appendix G: "bith" = mistake
13. Fig 6D/E: what is the color code orange/blue: before versus after training, or cluster 1 versus 2?
14. Fix typos in the legend of Fig 8 ("bottom right:)" / missing dot before "Model" / etc.)
15. Checklist: answer N/A would be more appropriate than YES for questions that don't apply

**Limitations:**

Limitations: The authors properly provide the limitations of their work

Societal impact: The authors don't discuss the potential negative societal impact of the work, which is fine given the nature of the topic

**Strengths And Weaknesses:**

Strengths (from most important to least important):
1. The paper provides a significant theoretical advancement to the literature on balanced networks, and suits the NeurIPS conference well
2. The work is very well related to the existing literature
3. The setup of the paper generalizes well to other transfer functions
4. Extremely pedagogical paper, especially when it comes to explaining each mathematical variable and the mathematical derivations (even too much pedagogical in section 3? You could probably move some steps to the supplementary material if you want to "gain" some space)
5. Numerical simulations are detailed and parameters are provided, which favors reproducibility
6. Interesting ideas are provided for follow-up work (low-rank networks / taking into account energetic and biophysical constraints)

Weaknesses (from most important to least important):
1. The "depth of balance" / "empirical balance" is an interesting notion (which seems new), but I would suggest using another notation than b. It is not clear what motivates the definition of empirical balance as mean recurrent coupling, and its link with the balance parameter b. Without a strong connection between the two notions, it is wrong to say that the higher mutual information after training was predicted by the theory, or that the network achieves tight balance after training. In the simulation, the parameter b is fixed at value 1...
2. The final part of the abstract about the network architecture is deceptive. It is expected that a balanced network (therefore dominated by inhibition) solving independent tasks will naturally form clusters, with mostly inhibitory intra-cluster connections and mixed weak inter-cluster connections (ideally, the two populations should probably not communicate here). What is observed in this simple context of independent tasks does not imply that "feedforward excitatory projections and local recurrent inhibition is a generic circuit motif for encoding and transmitting information". Additionally, there doesn't seem to be a preference for excitatory connections between clusters in Fig 6C and 6F
3. The theory lacks experimental predictions (one that comes to mind is the auditory cortex which encodes for different frequencies) and relation to existing experimental work
4. No code provided. Providing the code would help for even easier reproducibility. Note that anonymized links can be provided at the time of submission, by using an anonymized GitHub account for instance; see ICML guidelines: https://icml.cc/FAQ/submitted-code-anonymized / https://icml.cc/FAQ/github-links / https://icml.cc/Conferences/2022/StyleAuthorInstructions

---

> ### Author Response · Authors · 2022-08-04
> **clarification on 'empirical balance" & "excitatory projections  //added testable predictions / code provided**
>
> We thank the reviewer for their detailed comments and questions.  We will address the reviewer’s comments and questions in the following. First, would like to note that we have added two nontrivial tasks Figure 4E-F and a paragraph on testable prediction (line 304ff) and we have completely revised the manuscript.
>
> > Comment 1: The "depth of balance" / "empirical balance" is an interesting notion (which seems new), but I would suggest using another notation than b. It is not     clear what motivates the definition of empirical balance as mean     recurrent coupling, and its link with the balance parameter b.     Without a strong connection between the two notions, it is wrong to say that the higher mutual information after training was predicted by the theory, or that the network achieves tight balance after training. In the simulation, the parameter b is fixed at value 1.
>
> We clarified the notion of ‘empirical tightness of balance’ and also denote it by \hat b instead of b for improved clarity. We want to stress that the definition of empirical tightness of balance is consistent with the definition in our analytical work where the mean of the recurrent weights was chosen to be $-b J_0/N$, which also results in $ b=|1/N\sum_i\sum_j J_{ij}|$ in expectation. Thus, there is a direct relationship: Both \hat b and $b$ quantify strong the  mean inhibitory recurrent coupling is and, thus, how well networks can track high-frequency input.
> Further, we want to stress that \hat b is not fixed during training and b=1 is just the initialization of the network.
> We added in the main text (line 203ff):
> “This is confirmed by measuring the empirical tightness of balance $\hat b$ in the connectivity, which we quantify simply by the absolute value of the mean of the sum of incoming recurrent weights into each neuron $\hat b=|1/N\sum_i\sum_j J_{ij}|$.
> This definition of empirical tightness of balance $\hat b$  is consistent with the definition in our analytical work where the mean of the recurrent weights was chosen to be $-b J_0/N$ and thus $b J_0=|1/N\sum_i\sum_j J_{ij}|$ in expectation.
> We find that the trained networks that are initialized with $b=1$ become progressively more tightly balanced (Figure 3C) over training epochs, as can be seen by a growing $\hat b$.”
>
> > Comment: The final part of the abstract about the network architecture is deceptive. It is expected that a balanced network (therefore dominated by inhibition) solving independent tasks will naturally form clusters, with mostly     inhibitory intra-cluster connections and mixed weak inter-cluster connections (ideally, the two populations should probably not     communicate here). What is observed in this simple context of independent tasks does not imply that "feedforward excitatory     projections and local recurrent inhibition is a generic circuit     motif for encoding and transmitting information". Additionally,     there doesn't seem to be a preference for excitatory connections     between clusters in Fig 6C and 6F.
>
> Thank you for allowing us to clarify this: We did not want to claim, that the recurrent weights between clusters become excitatory over time, but that the external input weights become more excitatory over time. We added an inset in figure 3c to show that the input weights become more excitatory throughout training, while the recurrent weights become more inhibitory, consistent with the fact that tightness of balance b in Eq 1 both multiplies the positive external input I(t) and the inhibitory recurrent weight matrix.
>
>
> > Comment: The theory lacks experimental predictions (one that comes to mind is the auditory cortex which     encodes for different frequencies) and relation to existing     experimental work
>
> We thank the reviewer for pointing out that gap, and added the following paragraph to the discussion to cover both predictions and relations to experimental work:
> A testable hypothesis based on our findings is that changing the tightness of balance, e.g. by pharmacological manipulation or by a genetic knockout that affects net recurrent inhibitory strength (without generating runaway activity or pathological network states), would also affect the high-frequency stimulus encoding and information transmission. Specifically, we predict that stronger effective recurrent weights in conjunction with stronger excitatory inputs would improve both the high-frequency encoding and the mutual information rate between a stimulus and the population response. Conversely, when weakening recurrent inhibition and external input strength, we predict an impaired high-frequency encoding and a lower mutual information rate. Such predictions could not only be tested in vivo, but also by in vitro experiments, where the number of recurrent synapses can be manipulated.

---

> > ### Comment · Reviewer_H6Fq · 2022-08-08
> > **Response**
> >
> > Thank you for the answer.
> >
> > ### Answer to comment 1
> > The similarity between the notions of theoretical and empirical balance is now clear, thanks to the answer by the author. I was just wondering about the sentence "b is initialized at value 1" - I understand then that parameter b is allowed to change? I didn't see anything (figure or comment) about the value of b increasing from its initial value with the optimization process.
> >
> > ### Answer to comment 2
> > Thank you for the clarification. I still wonder why "feedforward excitatory projections" wouldn't correspond to the inter-cluster connections (although I assume that it is not so clear here as there is no notion of feedforward/feedback between your clusters as there is no hierarchy between them). What you instead assume is that it corresponds to the input from the external world.
> >
> > ### Answer to comment 3
> > I am happy with the interesting experimental predictions provided by the authors.
> >
> > I am keeping by evaluation as before (6 with confidence 4) and will be happy to reconsider it based on your answer to these last comments.

---

> > > ### Author Response · Authors · 2022-08-08
> > > **further clarification & slightly updated new version**
> > >
> > > We thank the reviewer for the answers and have a few additional clarifications:
> > >
> > > Answer to comment 1:
> > > ------
> > > Indeed, the sentence "b is initialized at value 1" in the training part can be misinterpreted, creating the illusion that b is still a parameter. Therefore, we have now written (line 190ff):
> > >
> > > "We initialize small-sized recurrent networks ($N=100$) that follow the dynamics of Eq. 1 in a loosely balanced state, with recurrent weights drawn i.i.d. from a Gaussian $J_{ij}\sim \mathcal{N}(-1/N,\,g^2/N)$ and input weights $w^{\textnormal{in}}_i$ and output weights  $w^{\textnormal{out}}_i$ are drawn from a Gaussian distribution with a positive mean. This initialization would correspond to  $b=1$ in our theoretically studied model (Eq.~1)."
> > >
> > > Answer to comment 2:
> > > ------
> > > > Thank you for the clarification. I still wonder why "feedforward excitatory projections" wouldn't correspond to the inter-cluster connections (although I assume that it is not so clear here as there is no notion of feedforward/feedback between your clusters as there is no hierarchy between them). What you instead assume is that it corresponds to the input from the external world.
> > >
> > > That is correct, we are only making a statement about the external input here (which in the literature is often assumed to be feedforward excitatory synaptic projections).
> > > To further clarify this, we have now changed the wording in the abstract (line 17):
> > >
> > > "Our findings suggest that feedforward excitatory input and local recurrent inhibition--as observed in many biological circuits--is a generic circuit motif for encoding and transmitting time-varying information in recurrent neural circuits."
> > >
> > > We are open to further changes to avoid any confusion, and want to thank the reviewers for their remarks that we feel have significantly improved the clarity of the manuscript.

---

### Official Review · Reviewer_HmY8 · 2022-07-12

**Rating:** 6
**Confidence:** 3
**Soundness:** 3 good
**Presentation:** 3 good
**Contribution:** 3 good

**Summary:**

This paper investigates the encoding of temporally-varying signals by neuronal populations using theory, simulation, and neural network approaches.   The main analytical findings are: (1)  non-stationary dynamic mean-field theory shows that a tighter balance between excitation and inhibition in the recurrent network allows better encoding of higher-frequency signals; (2) information-theoretic analysis shows that tight E-I balance maximizes the mutual information between the input and output. These analytical results are confirmed by simulations. Furthermore, the authors show that an autoencoder network trained to encode temporarily-varying signals exhibit strong local inhibition, indicating tight E-I balance. Thus, this work illuminates the potential functional advantages of local recurrent inhibition and tight E-I balance in the encoding of temporally-varying signals.

**Questions:**

Issues discussed in the weaknesses should be addressed.

**Limitations:**

The authors addressed the limitations of the approach but it would be helpful if they also address the concern about whether the implications of this work can be generalized to networks that perform a meaningful computation. There is no negative societal impact.

**Strengths And Weaknesses:**

Strengths:
The work is substantial and addresses the stated problem with multiple approaches from multiple perspectives. The analytical contribution and the autoencoder are novel.  The paper also provides intuitive insights as to why and how tight E-I balance in such a network might enhance information encoding and transmission.

Weaknesses:
The studied network primarily uses a neuronal population to encode the input time-varying signals rather than performing any meaningful information processing. It is not clear what representations those neurons assume. That is,  all those neurons appear to serve the simple function of a simple electrical wire, connecting the input and the output!   The only obvious function that the network performs seems to remove the noise and guard against the chaos caused by the imperfection of the neurons. Obviously, this cannot be the case.   Would the principle discussed in this paper generalize to  a network that actually performs some meaningful computation?

---

> ### Author Response · Authors · 2022-08-04
> **added two nontrivial tasks that show generality  // added figures on training dynamics, new tasks, different nonlinearities**
>
> We thank the reviewer for the helpful comments, which substantially improved the publications.
>
> > Comment: Weaknesses: The studied network primarily uses a neuronal population to encode the input time-varying signals, rather than performing any meaningful information processing.
>
> We now added two nontrivial tasks to demonstrate that the core prediction of our theory -  a higher information rate in tightly balanced networks, is also observed when networks are trained to perform nontrivial computations. Reliably encoding time-varying information into RNNs that are suspect to chaos and noise is a crucial prerequisite for performing subsequent information processing, so we deem a thorough theoretical treatment of that a scientifically important question.
>
>
> > Comment: It is not clear what representations those neurons assume.
>
> We now added figures that show that individual neuron responses can be very noisy and variable despite good performance. (Compare Figure 9 top middle to bottom left panel, same for Figure 10 and Figure 11). This shows that individual neurons' representation of the stimulus can still be highly unreliable while the population response encoding is very reliable in tightly balanced networks consistent with our non-stationary mean field theory.
>
> > Comment: That is, all those neurons appear to serve the simple function of a simple electrical wire, connecting the input and the output!
>
> A simple wire connecting input to output (e.i. Recurrent dynamics=0) would not solve the problem, because of the low-pass filtering property of the ‘wire’ considered here, that arises from the leak term in Eq. 1. We overcome this problem by strong recurrent inhibition that allows networks to respond to information encoded in the high-frequency component of the input despite recurrent chaos and added noise.
>
> > Comment: The only obvious function that the network performs seems to remove the noise and guard against the chaos caused by the imperfection of the neurons.
>
> The RNN actually does more. The network does not simple ‘average out’ the noise across neurons, but tightly balanced networks actively cancel the finite-size fluctuations in its mean $m$ by strong recurrent feedback that becomes faster (with timescale \tau/b) and stronger (with factor b)  with increasing tightness of balance $b$.
>
> > Comment: Obviously, this cannot be the case. Would the principle discussed in this paper generalize to a network that actually performs some meaningful computation?
>
> Indeed, the copy task is basic, we now added two nontrivial tasks to demonstrate that the core prediction of our theory -  a higher information rate in tightly balanced networks is also observed when networks are trained to perform nontrivial computations.
> Firstly, we added in Figure 4E, F and G  a continuous-time version of the XOR (the function f(x, y=|x-y|, which satisfies for x, y ∈{0,1} the binary XOR function). Thus, at any moment in time, the recurrent network has to generate the output f(x, y)=|x-y| of two continuous input signals (OU-processes), that have a high-frequency component. Consistent with our theory, the linearized network dynamics develops a negative real outlier eigenvalue (Fig 4F) throughout learning and the recurrent weight matrix shows develops balance subnetworks with strong inhibition (Fig 4F).
> Secondly, we trained RNNs to linearly transform an input vector $y_i(t)=\sum_{ij}A_{ij}I_j(t)$ at every moment in time. We observed that the RNNs develop a number of negative outlier eigenvalues and concomitantly the same number of tightly balanced inhibitory subnetworks that corresponds to the rank of the linear input-output transformation $A$ (See Fig 12 and appendix I for further details).
> In summary, this demonstrates that RNNs trained on nonlinear tasks which require rapid changes in the population firing rate based on high-frequency components of the input signal also develop a tightly balanced state throughout training. These are minimal example tasks designed to show that the theoretical link between ‘tightness of balance’ and information encoding our study establishes is also informative for the dynamics of task-optimized RNNs.
>
> > Comment:  The authors addressed the limitations of the approach, but it would be helpful if they also address the concern about whether the implications of this work can be generalized to networks that perform a meaningful computation. There is no negative societal impact.
>
> We hope our generalizations to nontrivial tasks above satisfied the concern, and we thank the reviewers for pushing us to demonstrate that tightly balanced networks can not only encode high-bandwidth information but also perform nontrivial computations. We are very open to further comments and suggestions.

---

> > ### Author Response · Authors · 2022-08-08
> > **Additional questions on nontrival examples?**
> >
> > As the author-reviewer discussion period is ending tomorrow, we wanted to politely ask if the main questions and concerns of the reviewer ("Would the principle discussed in this paper generalize to a network that actually performs some meaningful computation? ") were answered by the response to the reviewer's satisfaction.
> >
> > Just as a brief recap, we
> > * added two nontrivial tasks (Figure 4E-F + supplement) which demonstrate that the core prediction of our theory - an improved information encoding in tightly balanced networks despite chaos and noise is also observed when networks are trained to perform nontrivial computations that require high-frequencies.
> > * wrote a new section in the introduction that details the contribution of the paper.
> > * homogenized and clarified the mathematical notation
> >
> > We believe our responses have addressed all the questions raised in the review, but we would appreciate the opportunity to further clarify remaining open issues and respond to any further feedback. If there are no remaining questions, we want to thank again the reviewer for their helpful remarks that we feel have significantly improved the manuscript.

---

> > > ### Comment · Reviewer_HmY8 · 2022-08-09
> > > **Tasks better**
> > >
> > >
> > > I thank the authors for taking the effort to revise the manuscript and add the additional examples, particularly the XOR.  Because of this effort, I decided keep the original score despite of other reviewers' misgivings. The improved tasks are less trivial, but still not very compelling.  The point I wanted to make by the "wire analogy" is that if the function of the network is simply transmitting the signals, then it is no different from a wire functionally.

---

> > > > ### Author Response · Authors · 2022-08-09
> > > > **response on task**
> > > >
> > > > We thank the reviewer for taking the time to read through the revised manuscript. We agree that for a computer, a wire could do the task of transmitting a time-varying signal. However, we address the neuroscience challenge of how a signal can be reliably transmitted despite network chaos, noise and slow single neuron dynamics (originating from the leak in Eq. 1). We theoretically show how tight balanced networks resolve this challenge and demonstrate that trained networks seem to choose this solution.
> > > >
> > > > We would be very grateful for more concrete task requests, when the reviewer write, "The improved tasks are less trivial, but still not very compelling".

---

### Official Review · Reviewer_bJkp · 2022-07-15

**Rating:** 4
**Confidence:** 3
**Soundness:** 2 fair
**Presentation:** 2 fair
**Contribution:** 3 good

**Summary:**

The authors propose a non-stationary theory for recurrent neural network equations. In their framework, strong recurrent inhibition balances the inputs which in turn improves the mutual information rate between the inputs and ouputs of the network.

The contributions are the following :
- a theory that combine recurrent neural network equation, mean-field approximation and information theory,
- the relation between network balance and mutual information,
- numerical simulation accompanying the theory,
- additional simulation showing that training such a network on simulated date leads to more and more tightly balanced networks.



**Questions:**

- The theoretical results are hard to relate to the numerical results. The theoretical functions displayed are not explicit in the text eg what is the formula relating the mutual information I and the balance b.

- Equation (10) has been proved recently in
Komaee, A. (2020). Mutual information rate between stationary Gaussian processes. Results in Applied Mathematics, 7, 100107.
It holds for stationary Gaussian process. Any references supporting your claim that it holds for non-stationary process ?

- l 246 : Even if you work in somehow more general context, there is not really anything surprising about the fact that minimizing the MSE increases the mutual information, see
Guo, D., Shamai, S., & Verdú, S. (2005). Mutual information and minimum mean-square error in Gaussian channels. IEEE transactions on information theory, 51(4), 1261-1282.

- l 315 : As such, the manuscript is not convincing (I am not saying that it is wrong and I encourage the authors to continue their effort), why do you think your result holds under more general settings ?

Figure 1 A,B,C and Figure 3 D are not referred to in the main text.
Figure texts are too small.

Notations :
- coherence until section 3 then x and y are introduced as some mute variables introducing extra burden. I would advise to use x and y along the manuscript or I and \nu ...

Typos :
- l 196 => remove "how"
- l 207 => "I(t) is grows by b" ?
- l 225 => "e.i."
- l 226 => "sqrt2"
- l 232 => remove "mean squared loss"
- l 233 => y = \hat x ??
- l 252 => "e.i."



**Limitations:**

yes

**Strengths And Weaknesses:**

Strengths :
- the combination of multiple theories is appealing,
- the paper combine theory and simulation and also the relation with classical NN training,

Weaknesses :
- The mathematical notation are not coherent along the manuscript, impairing the clearest understanding,
- The structure of the manuscript conflates theory and numerical results,
- Many coarse typos (sometimes impairing understanding),
- Unsupported mathematical claims.

---

> ### Author Response · Authors · 2022-08-04
> **added nontrivial tasks  //added testable predictions // clarification regarding MSE and mutual info // added figures on training dynamics, new tasks, different nonlinearities**
>
> We thank the reviewer for their mathematically insightful comments and questions.  We will address the reviewer’s comments and questions in the following. First, would like to note that we have added two nontrivial tasks (Figure 4E-F) and a paragraph on testable prediction (line 304ff) and we have completely revised the manuscript.
>
> > Comment: The theoretical results are hard to relate to the numerical results. The theoretical functions displayed are not explicit in the text e.g. what is the formula relating the mutual information I and the balance $b$.
>
> We use our non-stationary DMFT to reduce the network dynamics of N ordinary differential equations into four non-local partial differential equations for m, \tilde h, c, and r (Equations 3a, 9, 10, and 11) and which are evaluated to obtain the population rate $\nu(t)$. Then the MI is calculated based on $\nu(t)$ and by evaluation equation 13. The approximate MI rate is obtained from Equation 20.
>
> > Comment: Equation (10) has been proved recently in Komaee, A. (2020).     Mutual information rate between stationary Gaussian processes.     Results in Applied Mathematics, 7, 100107. It holds for stationary     Gaussian process. Any references supporting your claim that it holds     for non-stationary process ?
>
> Thank you for giving us the opportunity to clarify: We are actually considering here a stationary Gaussian process as input, what is non-stationary is our mean-field theory as the autocorrelation explicitly depends on two different time points, which is necessary for a mean-field treatment of recurrent networks that receive time-varying mean input (Also see https://arxiv.org/abs/2201.09916). We did not want to claim that Eq. 10 is a novel result by us, we assumed this is a well-known formula (used e.g. Davide Bernardi and Benjamin Lindner, https://doi.org/10.1152/jn.00354.2014 or in other form by F. Rieke , D. A. Bodnar and W. Bialek https://doi.org/10.1098/rspb.1995.0204).We clarified that now in the main text
>
>
> > l 246 : Even if you work in somehow more general context, there is not really anything surprising about the fact that     minimizing the MSE increases the mutual information, see Guo, D.,     Shamai, S., & Verdú, S. (2005). Mutual information and minimum     mean-square error in Gaussian channels. IEEE transactions on     information theory, 51(4), 1261-1282.
>
> Correct, but this is not our knowledge claim. We show that minimizing MSE makes networks more tightly balanced for time-dependent input. For clarification, we now write:
>
>
> “Consistent with our theoretical results, we find that the mutual information rate quantified in the Gaussian channel approximations grows over training epochs as the network becomes more tightly balanced (Fig 3F). Note that we did not train the network to maximize the mutual information rate, but to minimize the mean-squared error between target output and actual output, but the two are closely related~\cite{guo_mutual_2005}.”
>
>
>
> > Comment: l 315 : As such, the manuscript is not convincing (I am not     saying that it is wrong and I encourage the authors to continue     their effort), why do you think your result holds under more general     settings ?
>
> We now added two additional nontrivial training results to demonstrate that the core prediction of our theory -  a higher information rate in tightly balanced networks - is also observed when networks are trained to perform nontrivial computations. Moreover, we demonstrated in figures 9-11 our results from figure 3 for other activation functions. Of course information encoding in RNNs with noise and chaos is only one aspect of information processing, but it is for an important issue for both theoretical neuroscience and also has implications for widely used continuous-time RNNs (e.g. echo state networks), and so far a mathematical theory linking ‘tightness of balance’ to temporal information encoding was missing. So we maintain that this is an important contribution suitable to the NeurIPS audience.
>
> > Comment: Figure 1 A,B,C and Figure 3 D are not referred to in the main text. Figure texts are too small.
>
> Thank you, corrected.
>
> > Comment: Notation: coherence until section 3 then x and y are introduced as some     mute variables introducing extra burden. I would advise to use x and     y along the manuscript or I and \nu ...
>
> Thank you, we homogenized the notation and  performed a thorough proofread and grammar check.

---

> > ### Author Response · Authors · 2022-08-08
> > **Reviewer questions addressed?**
> >
> > As the author-reviewer discussion period is ending tomorrow, we wanted to politely ask if the main questions and concerns of the reviewer were answered by the response to the reviewer's satisfaction?
> >
> > Just as a brief recap, we have
> > * improved and homogenized the mathematical notation
> > * added further clarification regarding the references the reviewer mentioned
> > * added two nontrivial tasks (Figure 4E-F + supplement)
> > * wrote a new section in the introduction that details the contribution of the paper.
> >
> > We believe our responses have addressed all the questions raised in the review, but we would appreciate the opportunity to further clarify remaining issues and respond to any further feedback. If there are no remaining issues, we want to thank again the reviewer for their insightful remarks that we feel have significantly improved the manuscript.

---

> > > ### Comment · Reviewer_bJkp · 2022-08-09
> > > **thanks**
> > >
> > > I thank you for your answer. It clarified some aspects of your manuscript. However, after reading the other reviews and answers, I am not inclined in increasing my score. I think overall that the manuscript is targeting a too narrow audience, yet this could be fixed.

---

> > > > ### Author Response · Authors · 2022-08-09
> > > > **How to target a broader audicence?**
> > > >
> > > > We thank the reviewer for taking the time. We would be grateful for feedback on how to target a broader audience with this manuscript to further improve our work.

---

> > > > > ### Author Response · Authors · 2022-08-09
> > > > > **remaining concerns?**
> > > > >
> > > > > We appreciate the in-depth evaluation of our work, and for bringing up the idea to target a broader audience. It is an important point, and something valuable to investigate in the evolution of our future work.
> > > > >
> > > > > The work as it stands in our rebuttal addresses all critical comments. Please advise if there is any other cause for your negative recommendation of our work. According to the reviewer guidelines, "3: Reject" is appropriate for "a paper with technical flaws, weak evaluation, inadequate reproducibility and incompletely addressed ethical considerations.” In all fairness, we do not believe our work falls into this category.
> > > > >
> > > > > We have fixed or rebutted all raised "technical flaws", and "narrow audience" is not to be among the evaluation criteria. If there remains any "technical flaws" or "unsupported mathematical claims", we would be grateful for feedback so we can improve our work.

---

### Official Review · Reviewer_5Aa7 · 2022-07-16

**Rating:** 5
**Confidence:** 4
**Soundness:** 3 good
**Presentation:** 2 fair
**Contribution:** 1 poor

**Summary:**

The authors develop an extension of the Dynamic Mean Field Theory to non-stationary inputs. They then use this toolkit to study how the tight balance of excitation and inhibition in recurrently connected networks supports information encoding of time-varying inputs to the network. Specifically, they find that the more tightly balanced networks are, the more they can "track" high-frequency fluctuations in the input through an effective reduction in the time constant of integration. They also train (with BPTT) a vanilla RNN to "copy" the input signal and validate their theoretical results.


**Questions:**

Please refer to the critiques listed above!

**Strengths And Weaknesses:**

The authors report a technical extension of the Dynamical Mean Field Theory to non-stationary inputs and apply that to a "copy" task. The notion of preserving/maintaining external inputs to the system (amidst noise) is certainly of interest to the neuroscience community. More relevant questions perhaps include the ability of RNNs to "keep track of time," which the authors can potentially tackle with their method. It also seems like certain metrics like the approximate mutual information are easier to calculate in their framework.

However, my main critique of this paper is its inadequacy of empirical evaluation and scope. The "copy" task is quite basic, and it seems overkill to develop and test theoretical scaffolding. This is, in fact, confirmed by the ability of out-of-the-box RNNs to learn this without sophisticated training techniques/architectural choices. Also, since the authors care about neuroscientific relevance, it is worth pointing out that in most cases neural circuits carry out computations of interest beyond simple transmission. If the authors believe that the "copy" task is indeed deceptively hard, can they please offer an explanation? Recommending this paper for publication requires at least one (or two) non-trivial demonstrations.

There are some other questions remaining about the manuscript which I have tried to detail below.

a. It is mildly confusing that the time-varying input signal x(t) (L77) never features in the definition of I(t). Can the authors clarify?

b. L88 "$b$ is a parameter that multiplies both external input and recurrent coupling strength and thus regulates the tightness of balance". As per Eq. 1 and L79, this isn't the case. Though the authors present a modified formulation for $J_{ij}$ in L92. Can the authors clarify?

c. L29 "where synaptic interactions scale O(sqrt(N)) with network size N". Can the authors elucidate more?

d. L73 "Our findings have important implications for ...". It would be quite helpful to have a contributions section and lay this and other contributions of this paper in the Introduction. Determining this submission's uniqueness/technical novelty is certainly challenging and can be partially alleviated through better writing principles.

e. L96 "by rewriting Eq 5a as". This is referenced before it is defined. Breaks the reader's flow. Kindly reformulate.

f. In Fig. 3, it would also be good to show the eigenvalue spectrum as a function of training epochs. Additionally, the authors state (L268) that their theory only describes the behavior of very large networks, yet the RNN they train is small (N = 100).

g. Generally, the paper can also do with a thorough proofread and grammar check. I've only highlighted a few of these below:

L69 "to reliable transmitting a"

L259 "other biophysically features"

L294 "biophysical more detailed"

L132 "how in linear response a"

L160 "alphabet [26]. ."

L196 "neural networks how how the"

L237 "(Figure 3A."

L226 "g = sqrt2"

L238 "(Figure 3B."

L241 "(Figure 3C"

L255 ", e.i.,"

---

> ### Author Response · Authors · 2022-08-04
> **Added two nontrivial tasks // added figures on training dynamics, new tasks, different nonlinearities**
>
> We thank the reviewer for the thoughtful comments, which substantially improved the publications. As requested, we added two nontrivial tasks (Figure 4E-F + supplement), added several figures on the training dynamics of the eigenvalues, and performed a thorough proofread and grammar check.
> Here are our more detailed responses:
>
> Comment: "However, my main critique of this paper is its inadequacy of empirical evaluation and scope. The "copy" task is quite basic, and it seems overkill to develop and test theoretical scaffolding.
> Recommending this paper for publication requires at least one (or two) non-trivial demonstrations."
>
> Indeed, the copy task is basic, we now added two nontrivial tasks to demonstrate that the core prediction of our theory -  an improved information encoding in tightly balanced networks despite chaos and noise is also observed when networks are trained to perform nontrivial computations.
> Firstly, we added in Figure 4E, F, and G  a continuous-time version of the XOR (the function f(x, y=|x-y|, which satisfies for x, y ∈{0,1} the binary XOR function). Thus, at any moment in time, the recurrent network has to generate the output f(x, y)=|x-y| of two continuous input signals (OU-processes), that have a high-frequency component. Consistent with our theory, the linearized network dynamics develops a negative real outlier eigenvalue (Fig 4F) throughout learning and the recurrent weight matrix shows develops balance subnetworks with strong inhibition (Fig 4F).
> Secondly, we trained RNNs to linearly transform an input vector $y_i(t)=\sum_{ij}A_{ij}I_j(t)$ at every moment in time. We observed that the RNNs develop a number of negative outlier eigenvalues and concomitantly the same number of tightly balanced inhibitory subnetworks that corresponds to the rank of the linear input-output transformation $A$ (See Fig~12 and appendix I for further details).
> In summary, this demonstrates that RNNs trained on nonlinear tasks which require rapid changes in the population firing rate based on high-frequency components of the input signal also develop a tightly balanced state throughout training. These are minimal example tasks designed to show that the theoretical link between ‘tightness of balance’ and information encoding our study establishes is also informative for the dynamics of task-optimized RNNs.
>
> Finally, we thank the reviewer for the suggestions regarding “keeping track of time” which we found particularly interesting, and we’d be thrilled to address such a question in future work.
>
> > Comment: "a. It is mildly confusing that the time-varying input signal x(t) (L77) never features in the definition of I(t). Can the authors clarify?"
>
> Clarified now.
>
> > Comment: "b. is a parameter that multiplies both external input and recurrent coupling strength and thus regulates the tightness of balance". As per Eq. 1 and L79, this isn't the case. Though the authors present a modified formulation for in L92. Can the authors clarify?"
>
> Corrected now.
>
> > Comment: "c. L29 "where synaptic interactions scale O(sqrt(N)) with network size N". Can the authors elucidate more?"
>
> We rewrote: “In networks with very strong recurrent inhibition, where recurrent weights are  scaled as $1/\sqrt{N}$”
>
> > Comment: "d. L73 "Our findings have important implications for ...". It would be quite helpful to have a contributions section and lay this and other contributions of this paper in the Introduction. Determining this submission's uniqueness/technical novelty is certainly challenging and can be partially alleviated through better writing principles.
>
> Thanks, added in line 76ff.
>
> > Comment: "e. L96 "by rewriting Eq 5a as". This is referenced before it is defined. Breaks the reader's flow. Kindly reformulate."
>
> Thanks, rewritten.
>
> > Comment: "f. In Fig. 3, it would also be good to show the eigenvalue spectrum as a function of training epochs. Additionally, the authors state (L268) that their theory only describes the behavior of very large networks, yet the RNN they train is small (N = 100)."
>
> Great suggestion, we now show the minimum real eigenvalue as a function of training epochs in Fig. 9, 10, and 11.  If requested so by the reviewer, we can replace one panel in Fig 3 with the full eigenvalue spectrum as a function of training epochs (e.g. the test error). Moreover, we analyze the learning dynamics of the eigenvalues in the 2D task in figure 7 and we added a movie of the eigenvalue dynamics to the anonymous code repository: https://anonymous.4open.science/r/NeurIPS2022-5340
> There is a new section “Training dynamics of eigenvalues” that shows the movie.
>
> > Comment "g. Generally, the paper can also do with a thorough proofread and grammar check. I've only highlighted a few of these below:"
>
> Done. We thank you again and would be very open to further questions and suggestions.

---

> > ### Author Response · Authors · 2022-08-08
> > **non-trivial demonstrations**
> >
> > As the author-reviewer discussion period is ending soon, we wanted to politely ask if the main concern of the reviewer ("Recommending this paper for publication requires at least one (or two) non-trivial demonstrations.") were answered to satisfactory. Just as a brief recap, we added two nontrivial tasks (Figure 4E-F + supplement), added several figures on the training dynamics of the eigenvalues and also wrote a new section in the introduction that details the contribution of the paper. Any further feedback would be appreciated.

---

> > > ### Comment · Reviewer_5Aa7 · 2022-08-09
> > > **Follow up experiments improve the paper!**
> > >
> > > Dear authors,
> > >
> > > Thanks for your effort in answering all of the questions above, and for newer experiments. Certainly Fig. 4 is very interesting and seems to corroborate some known facts about the emergence of specialized sub networks. Going through the paper again, I wasn't sure if you included trainable feed forward projection weights? (especially for the multiple channel OU inputs).
> > >
> > > One minor suggestion. When listing the contributions, its usually not preferred to include "analyses". Some of these bullets seem redundant to me. As far as I can tell, the non-stationary MFT is the key contribution. Rest are supporting proof-of-concepts.
> > >
> > > Emphasizing the "so-what?" factor of this theory can further help the manuscript.
> > >
> > > There are still several typos: for ex:
> > > L79 "between ti balance"
> > > Fig. 4 caption for subpanel (H) is labelled as (F)
> > >
> > > I am improving my score for this submission to a "borderline accept". My hesitation in strongly recommending this paper for acceptance stems from the fact that (though improved) the tasks used are fairly straightforward. The testable hypothesis emerging from this theory seem a bit stretched (given that the training algorithms used in this study are still non-biological, and hence these emergent properties might be quite different in neural systems).
> > >
> > > But, in general, I do commend the authors' efforts and wish them the best of luck!

---

> > > > ### Author Response · Authors · 2022-08-09
> > > > **further clarification on training & experimentally testable prediction**
> > > >
> > > > We thank the reviewer for reading through the revised manuscript and additional comments.
> > > >
> > > > We corrected the mentioned typos.
> > > >
> > > > Moreover, we state in line 195, that we train input weights, output weights and recurrent weights using backpropagation through time with ADAM optimizer with standard hyperparameters. This is also seen in the code that we provided.
> > > >
> > > > If there is a way to make that more clear, we'd appreciate the feedback.
> > > >
> > > > Finally, we want to stress that the core claim of our work  - an improved information encoding in tightly balanced networks despite chaos and noise - is not relying on any particular training scheme. The experimentally testable prediction that we make in line 305 is not relying on any biological or non-biological training algorithm, but is a statement about how the information encoding depends on tightness of balance:
> > > >
> > > > > A testable hypothesis based on our findings is that changing the tightness of balance, e.g. by pharmacological manipulation or by a genetic knockout that affects net recurrent inhibitory strength (without generating runaway activity or pathological network states), would also affect the high-frequency stimulus encoding and information transmission. Specifically, we predict that stronger effective recurrent weights in conjunction with stronger excitatory inputs would improve both the high-frequency encoding and the mutual information rate between a stimulus and the population response. Conversely, when weakening recurrent inhibition and external input strength, we predict an impaired high-frequency encoding and a lower mutual information rate. Such predictions could not only be tested in vivo, but also by in vitro experiments, where the number of recurrent synapses can be manipulated \cite{barral_synaptic_2016}.

---

> > > > > ### Author Response · Authors · 2022-08-09
> > > > > **added two additional panels in figure 9**
> > > > >
> > > > > To make our point about the increasing excitatory input during training more clear, we added two new panels to figure 9. The subplot on the bottom middle-right shows that the average of the excitatory external input weights in increasing throughout training.
> > > > > the subplot on the bottom right shows that (similar to figure 3) the empirical balance (see main paper for definition), is increasing. Both these findings consistent with our theory, where it would correspond to an increase in the parameter b. We note that the figure might give the impression, that the recurrent weights are growing stronger than the input weights, however, what matters for the balance are the excitatory and inhibitory input currents. The recurrent inhibitory input currents are giving by $\sum J_{ij}\phi(h(t))\le \sum J_{ij})$ for the sigmoid $\phi$ shown in figure 9.

---

### Meta-Review · Area_Chair_mymA · 2022-08-26

**Recommendation:** Accept
**Confidence:** Less certain

**Metareview:**

In spite of a somewhat weak empirical evaluation, reviewers appreciate the novel combination of multiple theories on recurrent networks.

**Award:**

No

---

### Decision · Program_Chairs · 2022-09-14

Accept